# Astrocytic microdomains from mouse cortex gain molecular control over long-term information storage and memory retention

Beatrice Vignoli [1,2✉], Gabriele Sansevero[3], Manju Sasi[4], Roberto Rimondini [5], Robert Blum [4], Valerio Bonaldo[2], Emiliano Biasini [2], Spartaco Santi [6,7], Nicoletta Berardi[8], Bai Lu[9] & Marco Canossa [2✉]

Memory consolidation requires astrocytic microdomains for protein recycling; but whether this lays a mechanistic foundation for long-term information storage remains enigmatic. Here we demonstrate that persistent synaptic strengthening invited astrocytic microdomains to convert initially internalized (pro)-brain-derived neurotrophic factor (proBDNF) into active prodomain (BDNFpro) and mature BDNF (mBDNF) for synaptic re-use. While mBDNF activates TrkB, we uncovered a previously unsuspected function for the cleaved BDNFpro, which increases TrkB/SorCS2 receptor complex at post-synaptic sites. Astrocytic BDNFpro release reinforced TrkB phosphorylation to sustain long-term synaptic potentiation and to retain memory in the novel object recognition behavioral test. Thus, the switch from one inactive state to a multi-functional one of the proBDNF provides post-synaptic changes that survive the initial activation. This molecular asset confines local information storage in astrocytic microdomains to selectively support memory circuits.

[1] Department of Physics, University of Trento, 38123 Povo (TN), Italy. [2] Department of Cellular Computational and Integrative Biology (CIBIO), University of Trento, 38123 Povo (TN), Italy. [3] Neuroscience Institute, National Research Council (IN-CNR), 56100 Pisa, Italy. [4] Institute of Clinical Neurobiology and Department of Neurology, University Hospital Würzburg, 97078 Würzburg, Germany. [5] Department of Medical and Surgical Sciences (DIMEC), University of Bologna, 40126 Bologna, Italy. [6] Institute of Molecular Genetics "Luigi Luca Cavalli-Sforza", National Research Council of Italy, 40136 Bologna, Italy. [7] IRCCS, Istituto Ortopedico Rizzoli, 40136 Bologna, Italy. [8] Department of Neuroscience, Psychology, Drug Research, Child Health (NEUROFARBA), University of Florence, 50100 Florence, Italy. [9] School of Pharmaceutical Sciences, Tsinghua University, 100084 Beijing, China.
✉email: beatrice.vignoli@unitn.it; marco.canossa@unitn.it

The most intriguing aspect of long-term potentiation (LTP) is the molecular basis essential for its persistence, a suggestive hallmark for memory consolidation. Crick first tackled the hypothesis that long-term memory storage is centered on a self-sustained molecular adaptation, suggesting protein phosphorylation as a paradigm[1]. Later on, the discovery of in-side activation of CaMKII[2], confirmed the existence of "memory molecules". CaMKII is initially activated by $Ca^{2+}$-Calmodulin. Due to a mechanism of self-phosphorylation that makes the enzyme constitutively active, CaMKII no longer requires the presence of $Ca^{2+}$-Calmodulin for its functionality[3,4]. More recently, Kandel and co-workers reported on a different process by showing that prion-like proteins, aggregated in a self-perpetrating state are involved in the stabilization of long-lasting synaptic changes through the control of local protein synthesis[5]. Thus, a variety of proteins that no longer experiences the initial activation can maintain memory. In line with this rather simple but attractive model, the mechanism that converts a transient neuronal stimulation into a persistent synaptic signaling during LTP has been a long-standing question[6].

Neurons are exquisitely specialized for rapid electrical transmission of signals, but glial cells, which do not communicate with electrical impulses are ideal for participating in cognitive functions requiring long-term temporal regulation and broad spatial integration[7–9]. In the present study, we report that astrocytic membrane protrusions enwrapping synaptic terminals—called microdomains—are a previously unsuspected synaptic compartment for the confinement of molecular memory. Once stimulated, neurons serve synthesis and release of brain-derived neurotrophic factor (BDNF) into the extracellular space[10], a key step in long-term synaptic modification[11]. In addition, neurons provide secretion of the precursor protein (~32 kDa proBDNF), consisting of prodomain (~12 kDa BDNFpro) and mature protein (~14 kDa mBDNF)[12]. To improve our understanding of the mechanisms controlling the extracellular availability of BDNF isoforms, we have previously examined the fate of both proBDNF and mBDNF in cortical brain slices after θ-burst-stimulation (TBS), a well-established paradigm inducing both LTP and secretion of BDNF[13]. We showed that BDNF, which is newly synthesized in neurons after TBS, is secreted in its pro-form and is then selectively internalized in astrocytic microdomains via p75$^{NTR}$-mediated endocytosis[14–16], thereby restricting the availability of the pro-neurotrophin at neuron–astrocyte contacts. After internalization, the pro-neurotrophin can undergo a recycling process, endowing astrocytes with the ability to stabilize LTP and retain memory[16]. Here we demonstrated that astrocytic microdomains coordinate the recycling process in subsequent steps, (i) the accumulation of BDNFpro and mBDNF proteolytic products likely generated by endocytic proBDNF processing; (ii) the vesicular storage and (iii) secretion of BDNFpro and mBDNF proteolytic products for synaptic re-use. While mBDNF commonly activate TrkB, astrocytic release of BDNFpro increases TrkB expression at the spine surface, which captures sufficient neurotrophin signaling for LTP maintenance. Hence, the persistence in synaptic strength is due to both an additional supply of mBDNF from astrocytic microdomains and to an increase in the post-synaptic response to neurotrophin by the pro-domain. Astrocytic release of BDNFpro, ultimately provides the molecular basis for retaining memory in the novel object recognition behavioral test. Thus, neurons and glia are associated by neurotrophins in functional memory units, which build reinforcing cellular and molecular loops enabling a persistent strengthening of the synapse and memory consolidation.

## Results

### Expression of BDNFpro in astrocytes.
We have previously reported that cortical layer II/III astrocytes support clearing and recycling of proBDNF[14,16,17] to sustain TrkB signaling and LTP maintenance in perirhinal cortex[16]. However, neither TrkB phosphorylation[18] nor the late-phase LTP[19,20] is directly regulated by proBDNF, suggesting that conversion of the inactive neurotrophin precursor to an active product might play a more direct role. We now ask whether these same astrocytes are proficient for proBDNF processing following LTP-inducing electrical stimulation.

Brain slices of control mice were prepared to examine the astrocytic origin of the proBDNF processing. Their perirhinal cortex was previously injected in layer II/III with adenoviral particles transducing green fluorescent protein (GFP) under the regulation of glial fibrillary acidic protein (GFAP) promoter (AAV-GFAP-GFP)[21]. Furthermore, in order to avoid the injection procedure, transgenic mice stably expressing GFP under the control of GFAP promoter (GFAP-GFP mice)[22] have been used. Slices were used for field stimulation delivering TBS and evoking LTP[13]. The expression of BDNFpro, expected from proBDNF proteolytic processing, was analyzed 10 min after stimulation by immunohistochemistry using an antibody (αBDNFpro) that specifically recognizes the furin cleaved C-terminal end of the prodomain (Fig. 1a[23]; and Supplementary Fig. 1). This epitope is unavailable in both intact proBDNF and mBDNF, providing that the antibody recognized the cleaved BDNFpro, leaving undetected cleavage-resistant proBDNF (proBDNF$^{CR}$) and mBDNF in the Western blot analysis (Fig. 1a). BDNFpro and GFP immunoreactivity was analyzed by confocal microscopy to appreciate the specific distribution of the cleaved prodomain in individual astrocytes (Fig. 1b and Supplementary Fig. 2a). GFP is a cytosolic protein whose fluorescence defines the astrocyte in its entire cytoplasmic extension. This is a feature that is ideal to achieve detection of BDNFpro in the astrocytic territorial volume. Spatial overlap of BDNFpro and GFP was analyzed in a series of confocal stacks by using colocalization analysis of the two signals (Supplementary Fig. 2c). To facilitate BDNFpro visualization in the astrocytic territorial volume, BDNFpro/GFP colocalization was reconstructed in z-stacks. We observed sharp BDNFpro/GFP colocalization signal, as detected by αBDNFpro, in astrocytes from TBS-slices (Fig. 1c and Supplementary Fig. 2b). Prodomain detection was observed in small proportion at the cell body and in greater proportion in the cell periphery mostly matching with highly ramified processes. In marked contrast, basal stimulation induced little BDNFpro/GFP colocalization signal in astrocytes (Fig. 1c and Supplementary Fig. 2b). For quantification analysis, we used Mander's overlap and measured the extent of co-occurrence[24] between the two fluorophores. By this analytical parameter, the proportion of BDNFpro/GFP colocalization was increased in the whole cell (Fig. 1c and Supplementary Fig. 2b) or branches subcellular regions (Fig. 1c) in TBS vs. basal conditions. Similar pattern of colocalization was observed using a specific antibody recognizing the mBDNF (mBDNF/GFP), the other resulting proteolytic product (Fig. 1d). An alternative colocalization parameter made similar results (Supplementary Fig. 2d).

Glial cells were reported to lack de novo proBDNF synthesis under physiological conditions[25], suggesting that BDNFpro and mBDNF detection in these cells originates from the proBDNF that was previously endocytosed and then cleaved. Preventing proBDNF internalization in astrocytes would then prevent the precursor processing and accordingly BDNFpro and mBDNF detection. This assumption was validated by experiments first conducted in slices treated with plasmin (100 nM), an enzyme responsible for the extracellular proteolysis of proBDNF to BDNFpro and mBDNF[26]. Upon proBDNF processing by the enzyme, astrocytes only express background levels of BDNFpro/GFP and mBDNF/GFP colocalizations (Fig. 1e). Thus, concomitant depletion of the precursor and

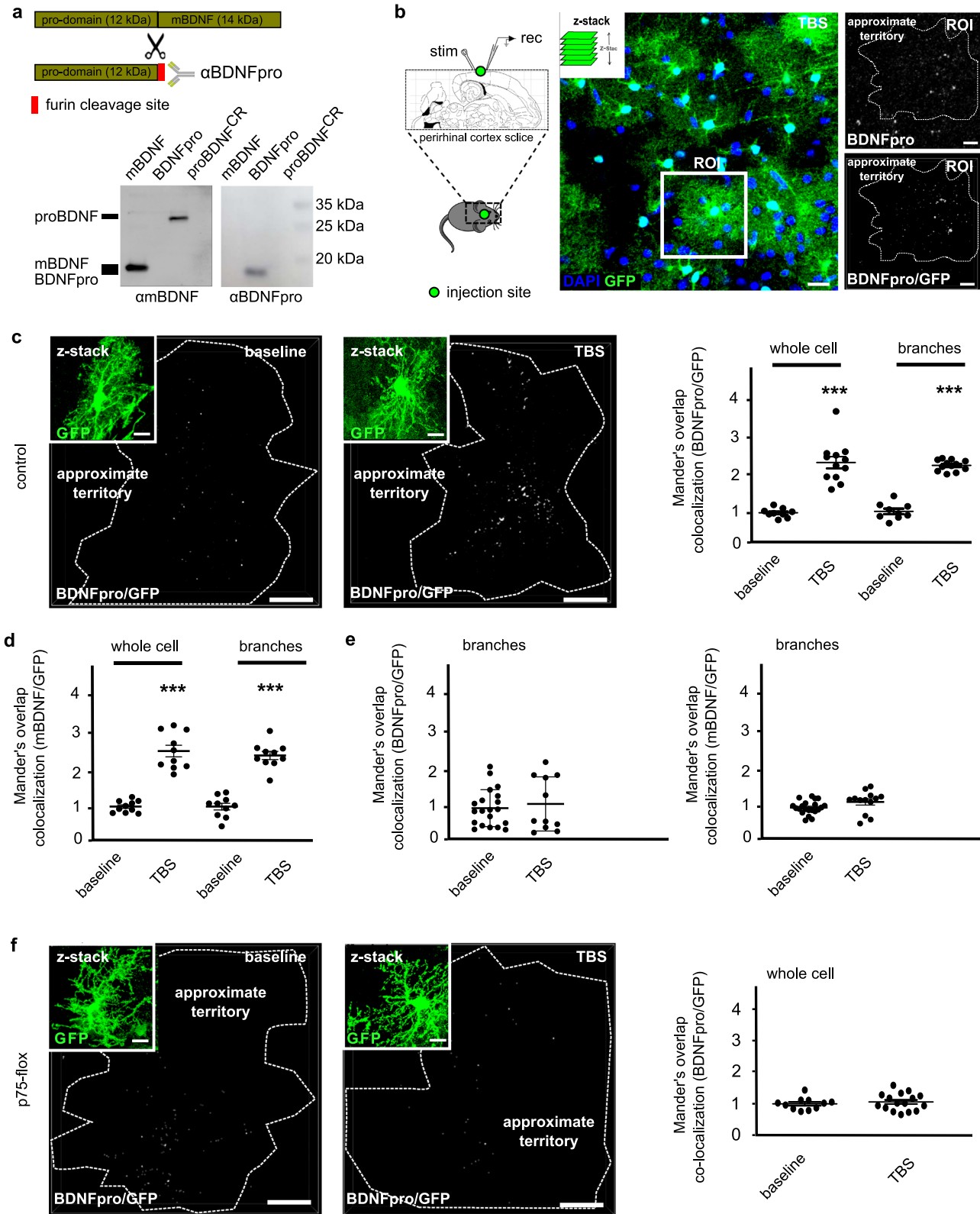

extracellular accumulation of its proteolytic products, suggests that proBDNF, but not BDNFpro and mBDNF, is selectively internalized by astrocytes. Second, we performed experiments in transgenic mice in which proBDNF uptake by astrocytes is prevented by deletion of p75[NTR], its carrier receptor[16]. These mice (from here on, p75-flox) were generated by crossing loxP-p75[NTR]-loxP mice[27] with GLAST-CreER[T2][28] and Rosa-CAGloxP-stop-

loxP(LSL)-R26R mice[29] (Supplementary Fig. 3a[30]). Tamoxifen administration by intraperitoneal injection into adult p75-flox mice induced p75[NTR] gene deletion, and expression of β-Galactosidase (β-Gal) reporter. Virtually, all β-Gal+ cells in the cortex showed the morphology of protoplasmic astrocytes (Supplementary Fig. 3b) and most of them were GFAP+ (81 ± 4%). Conversely, very few β-Gal+ cells expressed NeuN-neuronal marker (1 ± 0.7%),

**Fig. 1 BDNFpro expression in cortical astrocytes. a** Schematic representation of proBDNF precursor and cleaved BDNFpro domain. αBDNFpro antibody recognizes the furin cleavage site of the prodomain. Western blotting probing recombinant mBDNF, BDNFpro, and proBDNF$^{CR}$ with αBDNFpro and αmBDNF antibodies. **b** Cortical slices from control mice injected with AAV-GFAP-GFP virus were recorded and fixed 10 min after TBS for immunostaining. z-stack reconstruction shows astrocytes labeled by GFP. Magnification of a single stack from a region of interest (ROI) shows BDNFpro immunoreactivity and BDNFpro/GFP colocalization signal of one GFP-astrocyte delimited by an approximate territory (white dashed). Scale bars: 10 μm. **c** z-stack reconstruction of BDNFpro/GFP colocalization signals in astrocytes from baseline- and TBS-slices from control mice. The insets show GFP signal. BDNFpro/GFP colocalization was quantified in the whole cell and branches using Mander's overlap. ***$p < 0.001$ (unpaired $t$-test) ($n = 9$ cells, 4 slices, 3 mice for baseline; $n = 12$ cells, 4 slices, 3 mice for TBS). Scale bars: 10 μm. **d** mBDNF/GFP colocalization was quantified in the whole cell and branches using Mander's overlap. ***$p < 0.001$ (unpaired $t$-test) ($n = 10$ cells, 3 slices, 3 mice for baseline; $n = 10$ cells, 4 slices, 3 mice for TBS). **e** BDNFpro/GFP and mBDNF/GFP colocalizations in baseline and TBS-slices treated with plasmin were quantified in branches using Mander's overlap. ***$p < 0.001$ (unpaired $t$-test) ($n = 20$ cells, 3 slices, 3 mice for baseline BDNFpro/GFP; $n = 11$ cells, 3 slices, 3 mice for TBS BDNFpro/GFP; $n = 21$ cells, 3 slices, 3 mice for baseline mBDNF/GFP; $n = 13$ cells, 3 slices, 3 mice for TBS mBDNF/GFP). **f** z-stack reconstruction of BDNFpro/GFP colocalization signals in astrocytes from basal- and TBS-slices from tamoxifen-treated p75-flox mice. Insets show GFP signal. BDNFpro/GFP colocalization was quantified using Mander's overlap ($n = 11$ cells, 5 slices, 4 mice for baseline; $n = 16$ cells, 5 slices, 4 mice for TBS). Scale bars: 10 μm. Data are normalized to baseline and presented as mean ± SEM.

demonstrating that p75-flox mice allowed astrocytes selective targeting in the brain cortex in vivo. Immunoreactivity for BDNFpro was examined in slices from p75-flox mice 14 days post-tamoxifen (dptm). This latency ensured a significant depletion of p75$^{NTR}$ protein following recombination in astrocytes (Supplementary Fig. 3c). In p75$^{NTR}$-deficient cells, BDNFpro/GFP signal was hardly detectable in both TBS and basal stimulation (Fig. 1f). Given proBDNF, but not BDNFpro, is a ligand for p75$^{NTR}$[31], we conclude that p75$^{NTR}$-mediated proBDNF endocytosis likely feed astrocytes with a cleavable pool of proBDNF. These findings also reported on the specificity of our quantification analysis. Due to insufficient resolution of the confocal microscopy, BDNFpro/GFP colocalization signal could theoretically suffer contamination caused by synaptic BDNFpro. However, given p75$^{NTR}$-deficient astrocytes showed only background levels of BDNFpro/GFP co-localization, we estimated that the contribution of synaptic BDNFpro to the overall amount of astrocytic BDNFpro is minute, if not absent, and did not affect the relevance of our quantification analysis.

Collectively, our data suggest that proBDNF is selectively internalized in astrocytes and represents a substrate for proteolytic processing in cortical layer II/III astrocytes. Since, endocytic proBDNF undergoes recycling in this potentiating condition[16], our data suggest that peri-synaptic astrocytes might convert proBDNF into BDNFpro and mBDNF before routing to the secretory pathway.

**Localization of BDNFpro in astrocytic microdomains.** Cortical layer II/III astrocytes show a highly branched arborization and fine membrane extensions in the cell periphery[32]. The intricate ramifications of astrocytes allow them to tightly enwrap the synaptic terminals at organized peri-synaptic structures, the so-called microdomains[33]. Astrocytic microdomains can be structured into thick processes of micrometer scale (~10–15 μm$^2$) that host endoplasmic reticulum and mitochondria capable of generating inositol-3-phosphate-dependent Ca$^{2+}$ signals and thin organelle-free structures of submicron/nanometer-scale that fill the space between synapses[34–36]. We speculated that astrocytic proBDNF processing might be achieved on a rapid time scale at microdomains and, possibly, within the same storage compartments orchestrating the recycling process. Vesicular localization of the cleaved BDNFpro targeting astrocytic microdomains has been evaluated to be in line with our hypothesis.

Subcellular localization of BDNFpro was initially resolved in super-resolution by using structured illumination microscopy (SIM) (Fig. 2a). In TBS-slices from control mice, BDNFpro/GFP colocalization signal appeared as a punctate pattern dispersed into cell periphery of astrocytes. At higher magnification, BDNFpro/GFP colocalization was present in membrane

ramifications mostly shaped as finger-like extensions and flat lamellar sheaths (Fig. 2b), which were recognized to be astrocytic structures contacting the synapse. Quantification analysis confirmed that TBS induced an increase of the colocalization signal ($45 ± 6\%$ of the total BDNFpro/GFP puncta detected in astrocytes) in these specific structures. Given the nanometer-scale resolution of the SIM super-resolution microscopy (lateral res. 115 nm; axial res. 250 nm), our data suggest enrichment of the cleaved prodomain in high membranous elaborations of the cell periphery viewing the dimensions and typical morphology of microdomains.

To firmly assess this conclusion, we extended our investigation at ultra-structural levels[16]. Ultra-thin sections from TBS-slices were examined by transmission electron microscopy (EM) in pre-embedding experiments (Fig. 3a). At 70,000–100,000-fold magnification, immunogold labeling was observed using αBDNF-pro antibody (αBDNFpro-gold) in (i) vesicular structures at pre-synaptic terminals typically displaying clouds of synaptic vesicles opposed to post-synaptic density structures (Fig. 3b); (ii) post-synaptic terminals (Fig. 3b); and (iii) astrocytes giving rise to fine astrocytic processes in close proximity to the synapses (Fig. 3c). Consistent with our data that BDNFpro in astrocytes is generated from endocytic proBDNF processing in response to neuronal activity, TBS-slices showed a higher number of αBDNFpro-gold grains in astrocytes filling the space between synapses (Fig. 3d, e; $38 ± 7\%$ of the total gold particles detected in astrocytes) with respect to non-stimulated slices ($12 ± 4\%$ of the total gold particles detected in astrocytes). Most gold particles were organized in groups of many grains and were concentrated peri-synaptically ($46 ± 4\%$ of the total gold particles detected in astrocytes) in a membrane-delimited area of 230 nm radius surrounding synaptic contacts (Supplementary Fig. 4a). This distance represents the maximum of the astrocyte volume fraction when astrocytes cover dendritic spines and axonal boutons[37]. A similar enrichment was previously reported[16], using αmBDNF-gold (Supplementary Fig. 4b) or αproBDNF-gold (Supplementary Fig. 4c) potentially detecting both proBDNF and mBDNF isoforms or proBDNF and BDNFpro, respectively, suggesting for proBDNF processing in this area.

Lastly, we addressed the vesicular localization of BDNFpro. Endocytic proBDNF was shown to localize in vesicles expressing both p75$^{NTR}$[14,16] and Vamp2[38], a component of the core SNARE complex; thus indicating that intracellular storage of endocytic proBDNF in astrocytes involves vesicles equipped with the molecular machinery deputed to both endocytosis and exocytic fusion (recycling vesicles). In line with these findings, we discovered BDNFpro/GFP colocalization that overlapped with Vamp2/GFP colocalization in the astrocytic territory (Fig. 4a),

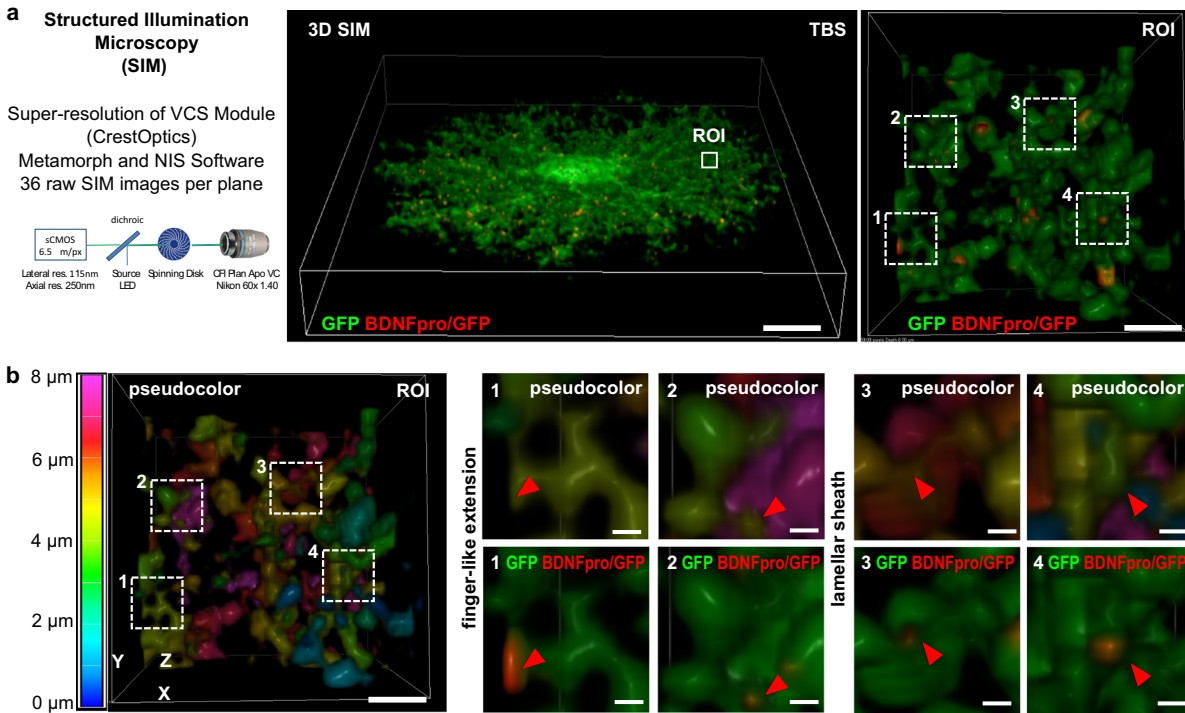

**Fig. 2 subcellular localization of BDNFpro. a** Graphical representation of the SIM super-resolution microscope. 3D-SIM image of a GFP-labeled astrocyte in a TBS-slice from control mice. Scale bar: 10 μm. Magnification of a ROI shows BDNFpro/GFP colocalization signal localized in fine membrane extensions of the cell periphery. Scale bar: 200 nm. **b** 3D-SIM image of the ROI in (**a**); z-axe is visualized in pseudocolor to facilitate microdomains identification. Scale bar: 200 nm. Magnification of microdomains characterized by the typical fingerlike extension (dashed squares 1 and 2) and flat lamellar sheath (dashed squares 3 and 4) are shown. BDNFpro/GFP colocalization is indicated (red arrowheads). Scale bars: 40 nm.

suggesting that the processing of the precursor takes place at recycling vesicles. Moreover, the vesicular localization of the prodomain in astrocytes was analyzed by BDNFpro/Vamp2 colocalization using SIM super-resolution microscopy. In TBS-slices, BDNFpro/Vamp2 colocalization signal appeared as a punctate pattern reminiscent of vesicular structures dispersed into cell periphery of astrocytes (Fig. 4b). Vesicular localization of BDNFpro was further observed at ultrastructural level (Fig. 4c, d); vesicles filled with BDNFpro-gold were occasionally seen at the astrocytic limiting membrane near synaptic endings. Similar vesicular localization was previously reported using αproBDNF-gold or αmBDNF-gold[16]; however, a reliable quantification of these storage organelles was inhibited by our pre-embedding procedure[16,39]. This procedure resulted in poor preservation of ultrastructural details in astrocytes, while ensuring the specificity of BDNFpro signal by preserving the antibody antigenicity. We conclude that vesicles containing BDNFpro are created at astrocytic microdomains in response to TBS.

**Astrocytic BDNFpro is sufficient for LTP maintenance.** A central hypothesis arising from proBDNF processing in recycling vesicles is that the resulting end products participate in gliotransmission, supplying astrocytes with a releasable source of active neurotrophins and enabling LTP maintenance. Conversion of proBDNF to mBDNF would typically satisfy this requirement. However, given that BDNFpro and mBDNF[16] share similar vesicular localization, we formulated the original hypothesis that BDNFpro individually participates to synaptic strengthening *via* the recycling process.

To assess the contribution of BDNFpro in LTP, we performed LTP-rescue experiments in conditional p75-flox mice as reported previously[16]. In these experiments astrocytes are incapable of proBDNF uptake, resulting in a short-lived potentiation that

declined to baseline about 140 min after TBS (Supplementary Fig. 5a). Conversely, expression of proBDNF (Supplementary Fig. 5b), but not cleavage resistant proBDNF[CR] (Supplementary Fig. 5c) in astrocytes restores LTP for the 180 min duration of the recording to levels exhibited by control littermates. An ectopic source of cleavable proBDNF in p75[NTR]-deficient astrocytes would then be assumed to replenish endocytic proBDNF and compensate for the lack of recycling in these cells, restoring the LTP deficits. Using this strategy, we expressed BDNFpro specifically in p75[NTR]-deficient astrocytes for LTP assessment.

First, we engineered a lentiviral construct (LV-BDNFpro[stop]) for BDNFpro expression (Supplementary Fig. 6a). In this construct, a loxP-GFP-STOP-loxP cassette allows GFP expression while preventing for BDNFpro expression. The GFP-STOP cassette is removed in the presence of Cre recombinase, finally resulting in GFP loss and activation of the BDNFpro transgene and Tomato reporter. We injected LV-BDNFpro[stop] into perirhinal cortices of p75-flox mice at 0 or 12 dptm. At 14 dptm (Supplementary Fig. 6b), cortical slices were prepared and stained for the astrocytic marker GFAP, the mouse reporter β-Gal, and the viral reporter Tomato. Most Tomato+ cells (98 ± 4%) were recombined astrocytes (GFAP+/β-Gal+) indicating the specificity of the lentiviral construct. In addition, we found that transduced astrocytes from both injection times expressed comparable levels of BDNFpro immunoreactivity, which was in the range of the one observed in TBS-slices from control littermates (Supplementary Fig. 6c). This is consistent with the expectation that virally transduced BDNFpro replaces the prodomain generated by endocytic proBDNF processing.

In parallel, slices were used for field recordings. While basal synaptic transmission (input–output; Supplementary Fig. 6d) and synaptic facilitation (paired-pulse facilitation; Supplementary Fig. 6e) were unaffected by LV-BDNFpro[stop] transduction, slices

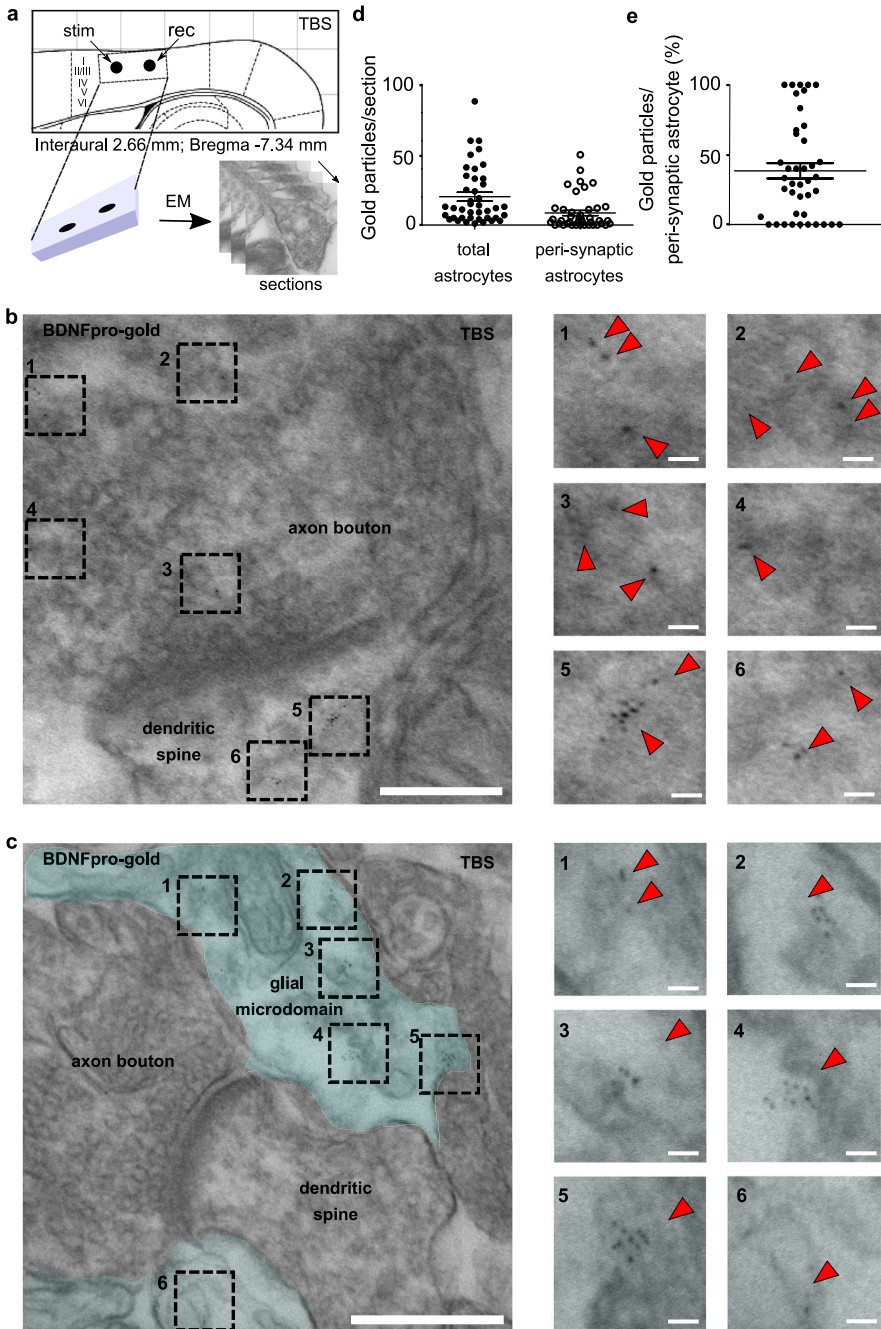

**Fig. 3 localization of BDNFpro in astrocytic microdomains. a** Experimental design linking field-potential with electron microscopy (EM) in layer II/III perirhinal cortex. TBS (10 min)-slices were dissected for EM processing. **b** Representative EM-image depicts BDNFpro-gold particles at axon bouton (dashed squares 1 to 4) and dendritic spine (dashed squares 5 and 6). Scale bar: 100 nm. Magnification indicates representative areas (dashed squares 1 to 6) in which gold particles (red arrowheads) localization is shown. Scale bars: 10 nm. **c** Representative EM-image depicts BDNFpro-gold particles (dashed squares 1 to 6) at astrocytic microdomains (light blue) Scale bar: 250 nm. Magnification indicates representative areas (dashed squares 1 to 6) in which gold particles (red arrowheads) localization is shown. Scale bars: 20 nm. **d** Dot plot depicts the number of BDNFpro-gold particles in whole astrocytes and peri-synaptic astrocytes counted *per* section ($n = 41$ sections, 5 slices, 3 mice). **e** Dot plot depicts the percentage of BDNFpro-gold particles at peri-synaptic astrocytes ($n = 41$ sections, 5 slices, 3 mice). Data are mean ± SEM.

from p75-flox mice injected with the virus showed to restore long-lasting LTP deficits (Fig. 5a and Supplementary Fig. 5d). Thus, supplying astrocytes with the sole BDNFpro expression compensates for the physiological presence of proBDNF uptake, processing, and final recycling of the proteolytic products. Prerequisite for this process is that virally transduced BDNFpro in astrocytes undergoes secretion enabling LTP maintenance. To assess this issue, we designed a lentiviral construct for specific

expression of Tetanus Toxin (TeTN) light-chain (LV-TeTN$^{stop}$) in astrocytes (Supplementary Fig. 6a). Sustained astrocytic expression of TeTN—a protease known to cleave the SNARE protein Vamp2[38]—is expected to inhibit the fusion of all secretory vesicles including those containing the neurotrophin[14]. We injected LV-TeTN$^{stop}$ together with LV-BDNFpro$^{stop}$ in perirhinal cortices of p75-flox mice the last day of tamoxifen treatment (Fig. 5b). At 14 dptm, cortical slices were used for LTP

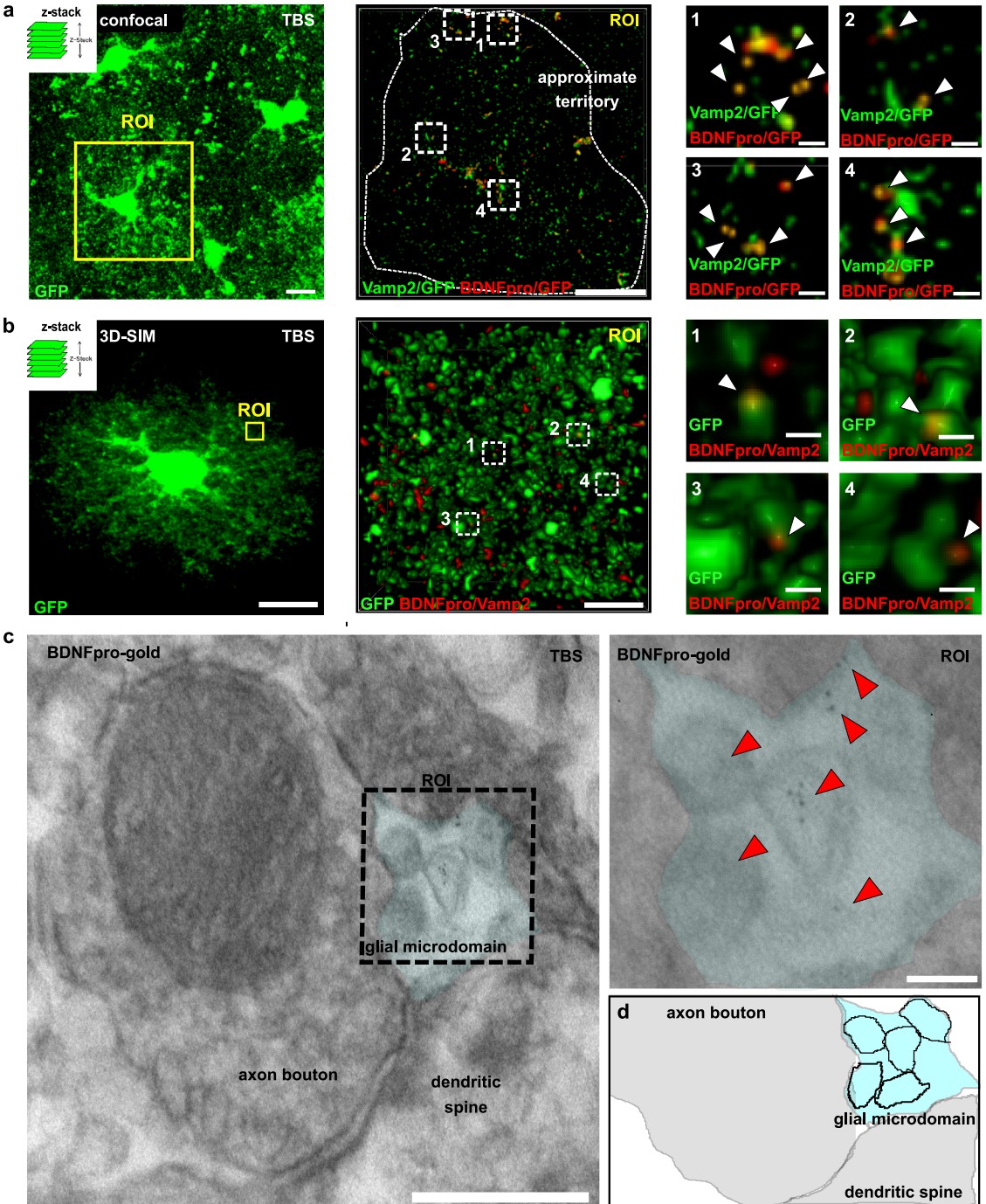

**Fig. 4 vesicular localization of BDNFpro. a** z-stack reconstruction shows astrocytes labeled by GFP. Cortical slices from control mice injected with AAV-GFAP-GFP virus were fixed 10 min after TBS and processed for immunostaining and confocal analysis. Scale bar: 10 μm. Magnification of a ROI shows one GFP-astrocyte delimited by an approximate territory (white dashed). Scale bar: 10 μm. BDNFpro/GFP and Vamp2/GFP co-localizations signals are shown. Magnification shows representative areas (dashed squares 1 to 4) in which BDNFpro/GFP and Vamp2/GFP signals overlap. Scale bars: 1 μm. **b** 3D-SIM image of a GFP-labeled astrocyte in a TBS-slice from control mice. Scale bar: 10 μm. Magnification of a ROI shows BDNFpro/Vamp2 colocalization signal. Scale bar: 500 nm. Magnification shows BDNFpro/Vamp2 colocalization signal in fine membrane extensions of the cell periphery (dashed squares 1 to 4). Scale bars: 50 nm. **c** EM image depicts BDNFpro-gold at astrocytic microdomains (light blue) surrounding an axon bouton. Scale bar: 100 nm. Magnification of the ROI shows gold particles (red arrowheads) in vesicular-like structures. Scale bar: 20 nm. **d** Digital reconstruction of the image in (**c**). Astrocytic vesicles (black boundary) are shown.

recordings. Slices from double-injected mice showed LTP that decayed to baseline 120–140 min in response to TBS; while co-injecting LV-GFP[stop] and LV-BDNFpro[stop] viruses into the contralateral hemisphere of the same mice showed to restore the LTP deficits to the levels exhibited by control littermates.

Thus, docking and fusion of Vamp2-secretory vesicles containing BDNFpro are required for LTP maintenance.

Whether BDNFpro is sufficient for rescuing LTP deficit was next investigated. Slices from p75-flox mice were perfused with exogenous BDNFpro (10 ng/ml) for 10 min (Fig. 5c). This timing

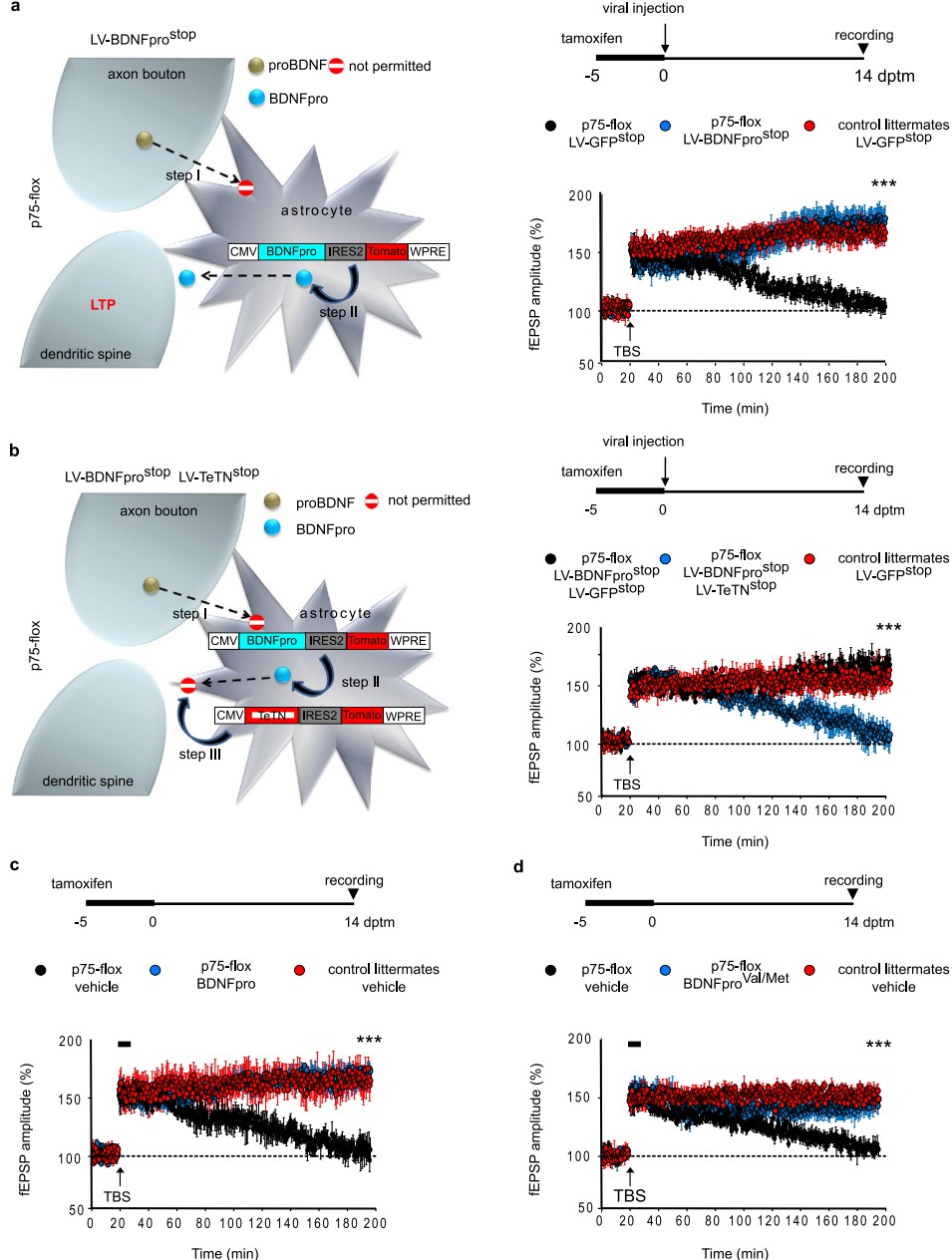

**Fig. 5 astrocytic BDNFpro secretion rescues LTP deficit in p75-flox mice. a** Schematic representation of the experimental design. Step I, deletion of p75[NTR] in astrocytes from tamoxifen-treated p75-flox mice precludes proBDNF transfer from neurons to astrocyte following TBS. Step II, LV-BDNFpro[stop] transduction replaces BDNFpro in astrocytes. Schematic representation of the experimental paradigms (right); mice were treated with tamoxifen (−5 to 0), injected with lentiviruses the last day of tamoxifen treatment (0 dptm) and finally recorded (14 dptm). LTP evoked in slices from p75-flox mice and control littermates injected with LV-GFP[stop] or LV-BDNFpro[stop] is shown. ***$p < 0.001$ (unpaired $t$-test) (p75-flox/LV-GFP[stop] 102.00 ± 1.85%, p75-flox/LV-BDNFpro[stop] 177.41 ± 10.74% and control littermates/LV-GFP[stop] 165.27 ± 2.24% fEPSP 180 min from TBS; $n = 10$ slices, 6 mice for p75-flox/LV-GFP[stop]; $n = 9$ slices, 6 mice for p75-flox/LV-BDNFpro[stop]; $n = 6$ slices, 4 mice for control littermates/LV-GFP[stop]). **b** Schematic representation of the experimental design. Step II, and I as in (**a**). Step III, LV-TeTN[stop] transduction in astrocytes prevents from BDNFpro release. Schematic representation of the experimental paradigm as in (**a**). LTP evoked in slices from p75-flox mice and control littermates injected with LV-GFP[stop] or co-injected with LV-GFP[stop]/LV-BDNFpro[stop] and LV-TeTN[stop]/LV-BDNFpro[stop] is shown. ***$p < 0.001$ (unpaired $t$-test) (p75-flox/LV-BDNFpro[stop]/LV-GFP[stop] 167.50 ± 8.78%, p75-flox/LV-BDNFpro[stop]/LV-TeTN[stop] 106.26 ± 2.47% and control littermates/LV-GFP[stop] 151.33 ± 7.39% fEPSP 180 min from TBS; $n = 8$ slices, 6 mice for p75-flox/LV-BDNFpro[stop]/LV-GFP[stop]; $n = 9$ slices, 7 mice for p75-flox/LV-TeTN[stop]/LV-BDNFpro[stop]; $n = 7$ slices, 4 mice control littermates/LV-GFP[stop]). **c** LTP evoked in slices from p75-flox mice and control littermates. Mice were treated with tamoxifen (−5 to 0) and recorded 14 dptm. Slices were perfused (18–28 min) with vehicle or exogenous BDNFpro. ***$p < 0.001$ (unpaired $t$-test) (p75-flox/vehicle 103.54 ± 3.43%, p75-flox/BDNFpro 171.09 ± 10.17% and control littermates/vehicle 162.81 ± 9.87% fEPSP 180 min from TBS; $n = 8$ slices, 4 mice for p75-flox/vehicle; $n = 7$ slices, 5 mice for p75-flox/BDNFpro; $n = 7$ slices, 5 mice for control littermates/vehicle). **d** LTP evoked as in (**c**). Slices were perfused (18–28 min) with vehicle or BDNFpro[Val/Met]. ***$p < 0.001$ (unpaired $t$-test) (p75-flox/vehicle 105.26 ± 1.59%, p75-flox/BDNFpro[Val/Met] 148.76 ± 8.69% and control littermates/vehicle 147.55 ± 3.55% fEPSP 180 min from TBS; $n = 8$ slices, 5 mice for p75-flox/vehicle; $n = 6$ slices, 4 mice for p75-flox/BDNFpro[Val/Met]; $n = 7$ slices, 5 mice for control littermates/vehicle). Data are presented as mean ± SEM.

was chosen as it correlates with the duration of proBDNF recycling in this cortical area[16]. Exogenous application of the BDNFpro fragment initiated 2 min before TBS and was maintained for an additional 8 min. We found that fEPSP was significantly restored in p75-flox mice receiving recombinant BDNFpro compared to vehicle-treated slices. Given that exogenous administration of mBDNF (10 ng/ml), but not cleavage-resistant proBDNF$^{CR}$ (20 ng/ml), similarly rescued LTP deficits (Supplementary Fig. 7a), our data indicate that proBDNF processing provides a mechanistic link between proBDNF clearing and subsequent recycling of the converted products, both possessing same individual ability to sustain LTP. Moreover, when TBS was omitted from the paradigm, BDNFpro application had no effect on fEPSP responses over a 180 min period of baseline test-stimulation (Supplementary Fig. 7b), indicating that BDNFpro requires TBS for participating in synaptic strengthening. The rescuing effect of recombinant BDNFpro is a transient phenomenon that saturates over time, as demonstrated by applying BDNFpro for 10 min at different time points after TBS (Supplementary Fig. 7c). The period of BDNFpro dependency ended about 110 min from TBS; after this time, application of the pro-fragment could no longer restore LTP deficit. Thus, BDNFpro is required for a limited time window to sustain LTP.

Collectively, our data indicate astrocytes to be as proficient for BDNFpro release, which fulfills the function to mediate the switch from early- to late-phase LTP.

**BDNFpro$^{Val/Met}$ preserves bioactivity.** A single-nucleotide polymorphism in the human BDNF gene results in valine (Val) to methionine (Met) substitution at codon 66 (Val66Met) in the prodomain region causing memory alteration in humans[40,41] and impaired synaptic strengthening in transgenic mice carrying the mutation[42]. By these premises, we investigated whether structural changes induced by this amino acid substitution might compromise BDNFpro function on LTP maintenance. Slices from p75-flox mice were perfused with recombinant BDNFpro carrying the genetic variation (BDNFpro$^{Val/Met}$) and subjected to TBS. We found that BDNFpro$^{Val/Met}$ (10 ng/ml) applied for 10 min in stimulated slices (Fig. 5d) retained the ability to restore LTP deficit. Thus, Val66Met substitution does not subtract bioactivity to BDNFpro in our experimental context.

**Astrocytic BDNFpro increases post-synaptic TrkB signaling.** BDNFpro is believed to be a ligand for the sortilin family member receptor SorCS2[31]. A potential function attributed to SorCS2 is to act as co-receptor for TrkB, assisting the assembling and targeting of TrkB/SorCS2 complex to post-synaptic membranes[43]. In this way, it recruits sufficient TrkB signaling for LTP maintenance. Once recycled by astrocytic microdomains, BDNFpro would then support aggregation and targeting of TrkB/SorCS2 complex at post-synaptic sites.

To assess this mechanistic issue, cultured cortical neurons expressing GFP were subjected to in situ proximity ligation assay (PLA)[43]. This experimental strategy requires PCR-amplification to hybridize fluorescent DNA probes coupled to secondary antibodies (αrabbit IgG and αgoat IgG) targeting αTrkB (rabbit) and αSorCS2 (goat) primary antibodies (Fig. 6a). Hybridization takes place only when probes are localized <40 nm apart, which reflects TrkB/SorCS2 clustering in our experimental context. In vehicle-treated neurons, basal PLA$^{TrkB/SorCS2}$-signal appeared as sparse fluorescent pattern (Fig. 6b). Fluorescent signal increased after treating neurons with exogenous BDNFpro (10 ng/ml) for 10 min. PLA$^{TrkB/SorCS2}$-signal was prominent in the plasma membrane of both cell body and dendritic processes, indicating a clear post-synaptic localization of TrkB/SorCS2 aggregates. Strikingly, PLA$^{TrkB/SorCS2}$-signal in

dendrites was predominately concentrated in proximity of spines, appearing as membrane protrusion extending from dendritic processes (Fig. 6c). Quantitative analysis confirmed the increase of PLA$^{TrkB/SorCS2}$-signal in BDNFpro-treated vs. vehicle-treated cultures (Fig. 6d). The specificity of BDNFpro treatment was assessed by pre-treating neurons for 20 min with αSorCS2 (20 μg/ml) (Fig. 6d), a blocking antibody that is known to prevent TrkB/SorCS2 complex formation[43]. In the presence of αSorCS2 pre-treatment, BDNFpro fragment could no longer exert its aggregating effect and PLA$^{TrkB/SorCS2}$-signal could not be seen (Fig. 6d). Moreover, when neurons were treated with recombinant mBDNF (10 ng/ml), or cleavage-resistant proBDNF$^{CR}$ (20 ng/ml), PLA$^{TrkB/SorCS2}$-signal had not changed significantly as compared to vehicle-treated neurons (Fig. 6d); thereby, within BDNF isoforms, TrkB/SorCS2 aggregation is a unique function of BDNFpro. PLA analysis was extended to cortical slices from control mice 10 min after basal- or TBS-stimulation (Fig. 6e). We measured in a quantitative analysis, PLA$^{TrkB/SorCS2}$ signal that colocalized with the postsynaptic marker NeuN (NeuN/PLA$^{TrkB/SorCS2}$) for cell body localization, and PSD95 (PSD95/PLA$^{TrkB/SorCS2}$) for spines localization at the stimulated areas. In TBS-slices, both NeuN/PLA$^{TrkB/SorCS2}$ and PSD95/PLA$^{TrkB/SorCS2}$ colocalizations had significantly increased with respect to baseline stimulation, indicating that TrkB and SorCS2 aggregate at post-synaptic sites in response to potentiating conditions. The astrocytic dependency on TrkB/SorCS2 aggregation was next examined in TBS-slices from p75-flox mice previously injected with control LV-GFP$^{stop}$ or LV-BDNFpro$^{stop}$ viruses. TBS-slices transduced with LV-BDNFpro$^{stop}$ showed higher levels of NeuN/PLA$^{TrkB/SorCS2}$ signals with respect to slices transduced with LV-GFP$^{stop}$ (Fig. 7a, b). Thus, BDNFpro sourced by astrocytes increases TrkB/SorCS2 aggregation at post-synaptic sites following LTP-inducing neuronal stimulation.

Finally, high-localized TrkB/SorCS2 aggregation would result in increased TrkB activity. To assess this possibility, we examined TrkB phosphorylation (pTrkB) by using α-phospho-TrkB (Tyr816) antibody (αpTrkB). Given that pTrkB is essentially a fraction of the total TrkB levels, we measured, in quantitative analysis, pTrkB that colocalizes with TrkB immunoreactivity. The benefit for this co-localization is related to the antibody specificity: only when pTrkB fluorescence co-localized with TrkB immunoreactivity was the signal considered a phosphorylated receptor. Moreover, using TrkB immunoreactivity as internal control ensured a better quantification of pTrkB reducing the variability between preparations. We found that the levels of pTrkB/TrkB colocalization had significantly increased in TBS-slices compared to non-stimulated slices from p75-flox mice injected with LV-BDNFpro$^{stop}$ or TBS-stimulated slices from control littermates injected with LV-GFP$^{stop}$ (Fig. 7c and Supplementary Fig. 8). Moreover, this increase is comparable to the one observed in TBS vs. baseline condition in control littermates (Fig. 7c). Thus, astrocytic BDNFpro recruits TrkB expression on adjacent spines for tight temporal, spatial, and stimulus-dependent TrkB phosphorylation.

**Astrocytic BDNFpro restores memory retention.** Substantial evidence suggests that the perirhinal cortex plays a critical role in familiarity-based object recognition memory[44,45].

To assess whether BDNFpro originated by astrocytes is involved in recognition memory in vivo, we injected LV-GFP$^{stop}$ or LV-BDNFpro$^{stop}$ viruses in perirhinal cortex of p75-flox mice the last day of tamoxifen treatment and performed the object recognition test (ORT) 14 dptm. Mice performed the ORT with a retention interval between the sample phase and the test phase of 10 min and 24 h (Fig. 8a). The discrimination index can vary between +1 and −1, where positive scores indicate more

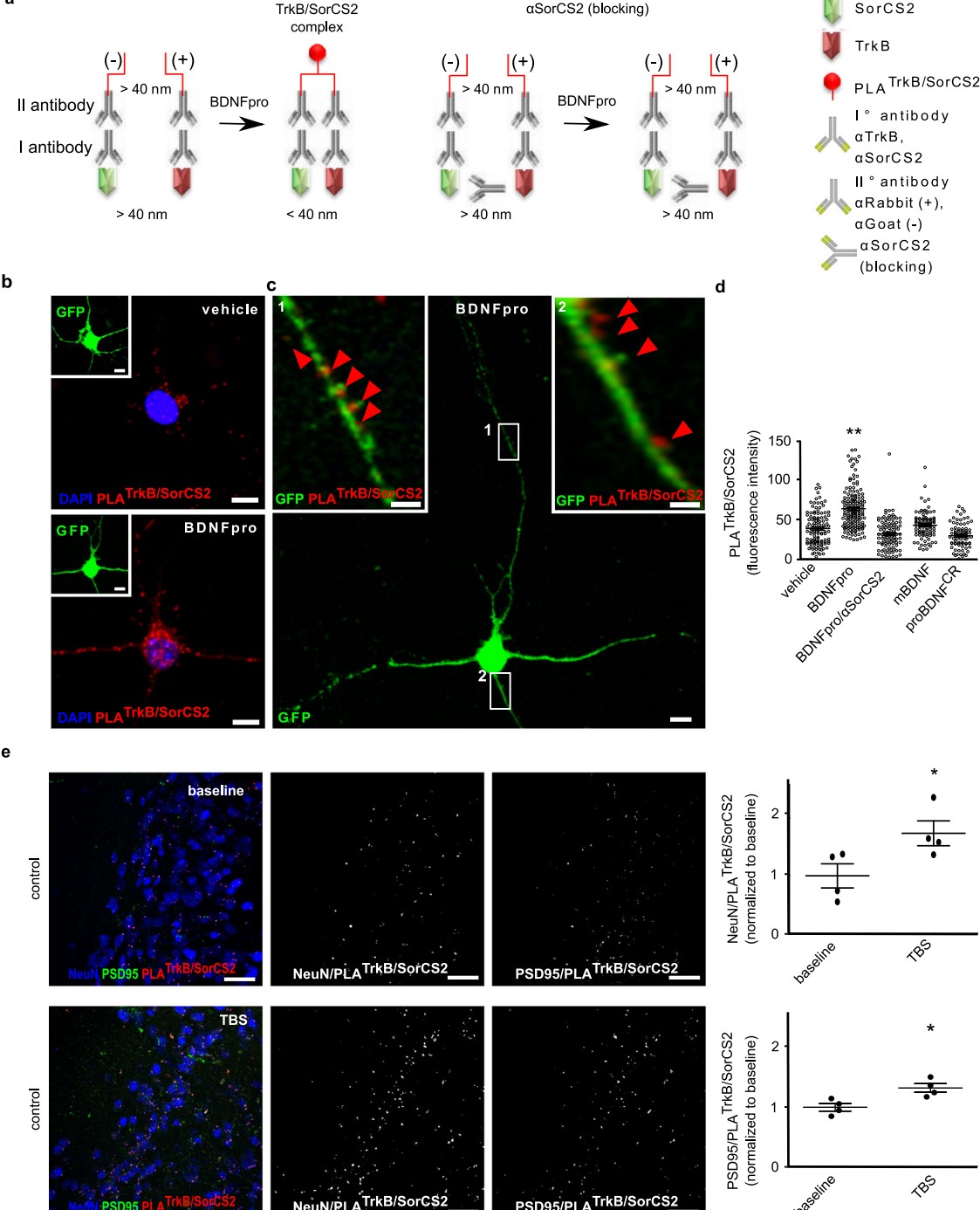

**Fig. 6 post-synaptic targeting of TrkB/SorCS2 complex. a** Schematic representation of the experimental design. Circular DNA probes (−) and (+) are coupled to II° antibody targeting αSorCS2 and αTrkB I° antibody. BDNFpro induces TrkB/SorCS2 complex formation (PLA$^{TrkB/SorCS2}$) that is prevented in the presence of αSorCS2 (blocking) antibody. **b** Panels show PLA$^{TrkB/SorCS2}$ signals in primary culture of cortical neurons treated with vehicle or BDNFpro. The insets show reference GFP-neurons. Scale bars: 5 μm. **c** Panels show a GFP-neuron treated with BDNFpro. Scale bar: 5 μm. Magnification of regions of interest 1 and 2 shows dendritic PLA$^{TrkB/SorCS2}$ localization (red arrowheads). Scale bar: 1 μm. **d** Quantification of PLA$^{TrkB/SorCS2}$ signal in cultured neurons treated with vehicle, BDNFpro (in presence or absence of αSorCS2), mBDNF or proBDNF$^{CR}$. Data are presented as mean ± SEM; **$p < 0.01$ (unpaired $t$-test) ($n = 111$ cells, 3 cultures for vehicle; $n = 152$ cells, 4 cultures for BDNFpro; $n = 98$ cells, 3 cultures for BDNFpro/αSorCS2; $n = 89$ cells, 3 cultures for mBDNF; $n = 79$ cells, 3 cultures for proBDNF$^{CR}$). **e** z-stack reconstruction showing NeuN, PSD95 and PLA$^{TrkB/SorCS2}$ signals in baseline and TBS slices. NeuN/PLA$^{TrkB/SorCS2}$ and PSD95/PLA$^{TrkB/SorCS2}$ colocalization signals are shown. Scale bars: 40 μm. NeuN/PLA$^{TrkB/SorCS2}$ and PSD95/PLA$^{TrkB/SorCS2}$ colocalization was quantified using Mander's overlap. Data are normalized to baseline and presented as mean ± SEM; *$p < 0.05$ (unpaired $t$-test) ($n = 4$ slices, 3 mice for each experimental condition).

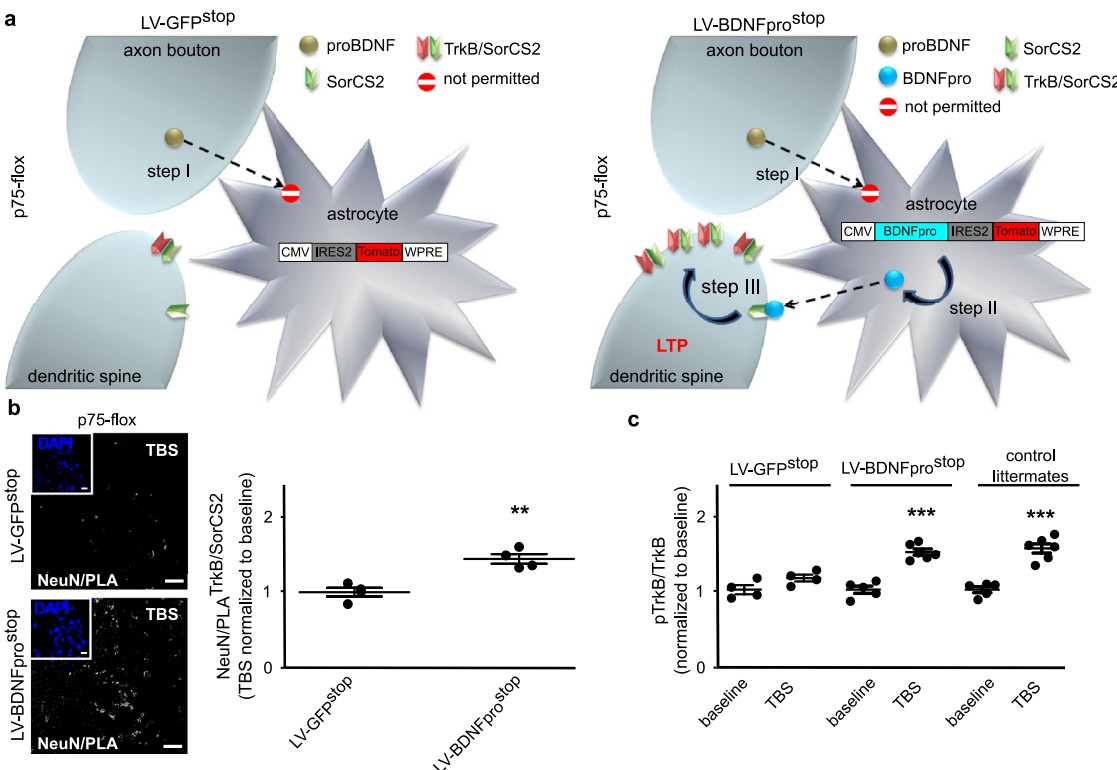

**Fig. 7 BDNFpro-induced TrkB/SorCS2 targeting. a** Schematic representation of the experimental design. Step I, deletion of p75[NTR] in astrocytes from tamoxifen-treated p75-flox mice precludes proBDNF transfer from neurons to astrocyte following TBS. Step II, LV-BDNFpro[stop] transduction replaces BDNFpro in astrocytes. Step III, astrocytic BDNFpro provides final increase of TrkB/SorCS2 complexes in dendritic spines and LTP maintenance. **b** z-stack reconstruction showing NeuN/PLA[TrkB/SorCS2] colocalization signal in TBS-slices from p75-flox mice transduced with LV-GFP[stop] or LV-BDNFpro[stop]. Scale bars: 40 μm. The insets show the field of analysis. Scale bars: 15 μm. NeuN/PLA[TrkB/SorCS2] colocalization was quantified using Mander's overlap. **$p < 0.01$ (unpaired $t$-test) ($n = 4$ slices, 3 mice for each experimental condition). **c** Dot plot shows quantification of pTrkB/TrkB colocalization in baseline- and TBS-slices from p75-flox mice transduced with LV-GFP[stop] or LV-BDNFpro[stop] and control littermates using Mander's overlap. ***$p < 0.001$ (unpaired $t$-test) ($n = 4$ slices, 3 mice for p75-flox mice/LV-GFP[stop]/TBS; $n = 4$ slices, 3 mice for p75-flox mice/LV-GFP[stop]/baseline; $n = 6$ slices, 4 mice for p75-flox mice/ LV-BDNFpro[stop]/TBS; $n = 5$ slices, 4 mice for p75-flox mice/LV-BDNFpro[stop]/baseline; $n = 5$ slices, 3 mice for control littermates/TBS; $n = 6$ slices, 3 mice for control littermates/baseline). Data are normalized to baseline and presented as mean ± SEM.

time spent with the novel object and a zero or negative score indicates equal or less time spent with the novel object. In the discrimination index (Fig. 8b) p75-flox mice injected with control virus showed positive values at 10 min time that decreased to almost zero at 24 h. Thus, when astrocytic proBDNF uptake was blocked, as in p75-flox mice, memory consolidation was prevented and mice showed a significant memory deficit in the ORT[16]. Strikingly, p75-flox mice injected with a virus transducing BDNFpro showed similar positive value at both 10 min and 24 h times, as for control littermates, indicating that astrocytic BDNFpro is sufficient for the consolidation of this type of memory. Exploration time at the sample phase was comparable between groups (Fig. 8c). Thus, BDNFpro in astrocytes could reverse the memory defect exhibited by p75-flox mice, demonstrating its necessity to memory consolidation in vivo.

## Discussion
At the micro/nanoscale, neural plasticity emerges as changes in the spatiotemporal pattern of activation of different synaptic components. While much attention has been given to pre- and post-synaptic sites for coordinating long-lasting synaptic strengthening[6,11], research in this regard has long time ignored the function of astrocytic microdomains. The close anatomical interface between pre- and post-synaptic neurons and astrocytes has been referred to as the synaptic triate[46] or tri-partite

synapse[47] in which astrocytic microdomains are proposed as regulatory units of neuron-glia interaction. Our work indicates these cellular structures to be specialized in prolonging synaptic potentiation by providing functional BDNFpro and mBDNF products, thereby increasing temporal, spatial, and stimulus-dependent neurotrophin availability at spines. BDNFpro and mBDNF reinforce TrkB signaling *via* adaptive molecular mechanisms that promote LTP maintenance and memory consolidation. This molecular functionalization process might involve individual or small groups of synapses covered by thin (nanometer scale) or thick (~10–15 μm²) microdomains, respectively[33,48,49]. It is therefore reasonable to assume that the spatial extent of BDNFpro localization at microdomains might be a constraining factor for the effect of these substructures in synaptic modifications. This spatially restricted mechanism of plasticity may finally participate in long-term memory engram formation[50] and memory capacity[51].

Astrocytic proBDNF is inactive in the precursor state and can be converted to active mBDNF in astrocytic microdomains. This inactive-to-active transition increases mBDNF availability exceeding the transient, activity-dependent neurotrophin secretion from neurons. This represents per se a positive cellular and molecular loop: (i) proBDNF transfer from neurons to astrocytes; (ii) processing and storage of the precursor; (iii) mBDNF transfer from astrocytes to neuron; that survives to LTP induction. However, our data provide further insight into this molecular

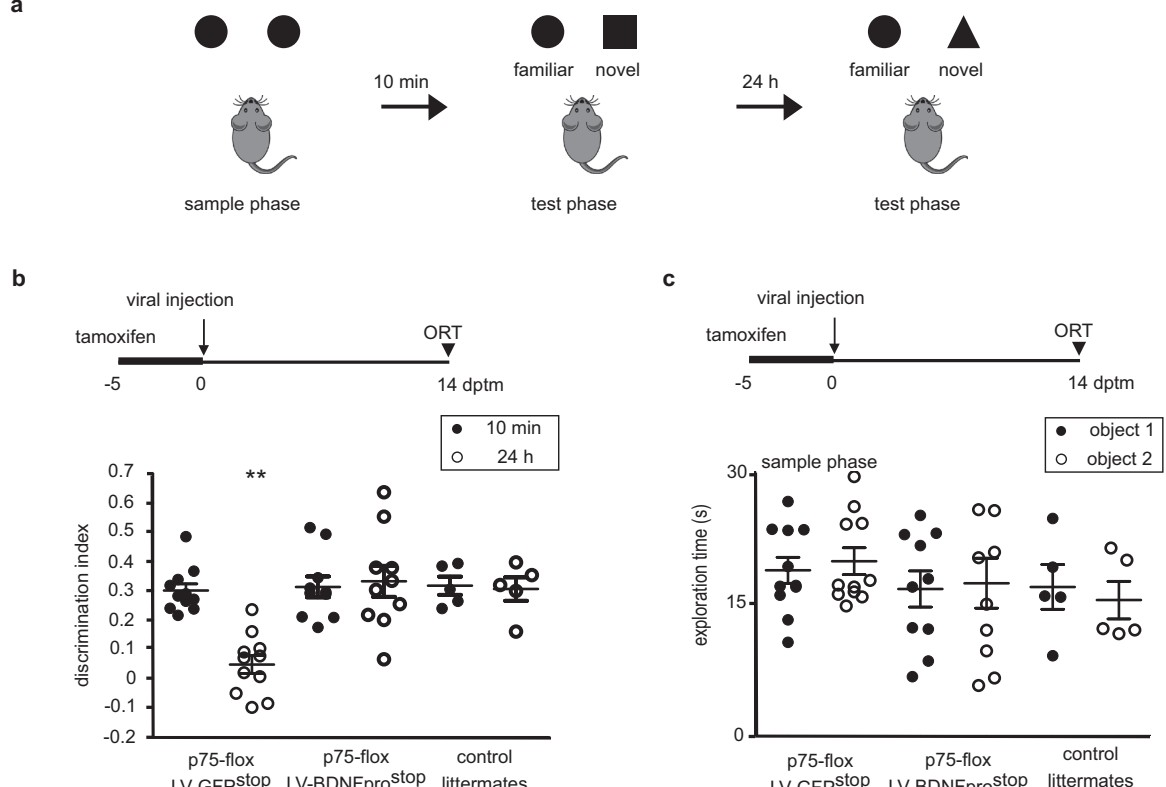

**Fig. 8 BDNFpro restores memory retention in p75-flox mice. a** Schematic diagram depicting the behavioral paradigm used for ORT. Mice were subjected to familiarization (sample phase) with two identical objects (circles). A test phase in which one familiar object (circle) is substituted with a novel one was performed after 10 min (square) and 24 h (triangle). **b** Schematic diagram depicting the experimental paradigm. p75-flox mice and control littermates treated with tamoxifen (−5 to 0) and injected with LV-GFP[stop] or LV-BDNFpro[stop] the last day of tamoxifen treatment (0 dptm) were subjected to ORT (14 dptm). Discrimination index is plotted against time interval between sample phase and test phases. **p < 0.01 (post hoc Holm–Sidak) (n = 11 mice for p75-flox/LV-GFP[stop]; n = 10 mice for p75-flox/LV-BDNFpro[stop]; n = 5 mice for control littermates). **c** Schematic diagram depicting the experimental paradigm as in (**b**). The dot plot shows mean exploration time of the familiar object and the novel object in the sample phase (n = 11 mice for p75-flox/LV-GFP[stop]; n = 10 mice for p75-flox/LV-BDNFpro[stop]; n = 5 mice for control littermates). Data are presented as mean ± SEM.

activation; not only mBDNF, but also the BDNFpro turned out to possess an independent function. Once generated, this byproduct participates in astrocytic recycling by inducing changes in TrkB levels (Fig. 7b) as well as in this receptor phosphorylation at post-synaptic sites (Fig. 7c). Thus, BDNFpro operates synaptic adaptations once LTP is expressed that are relevant for its later stabilization (Fig. 5a, c). Mechanistically, BDNFpro acts as an activator of TrkB/SorCS2 aggregation by spine targeting (Fig. 6c, e). This is a new physiological role attributed to the cleaved BDNFpro, which on behalf of its structural instability has been recently assigned to co-secretion with mBDNF[31,39] and assumed to possess independent physiological functions[23,52]. While previous data reported that exogenous application of proBDNF[12] or BDNFpro[52] enhances long-term depression (LTD)-inducing low frequency-stimulation in the hippocampus, we provide compelling evidence that in perirhinal cortex astrocytic proBDNF recycling and therefore astrocytic release of its proteolytic products BDNFpro and mBDNF do not participate in this form of facilitation[16]. This emphasizes the relevance to consider different brain regions and to identify the precise location of neurotrophin secretion at synaptic level. Although extremely challenging experimentally, these important issues must be addressed in order to clarify the current inconsistency of the involvement of neurotrophins in synaptic plasticity and processes related to memory[53].

Overall, the main question underlying astrocytic proBDNF processing can be further detailed: is the persistence in synaptic strength due to an additional supply of active mBDNF from astrocytic microdomains or is it due to an increase in the post-synaptic response to this neurotrophin by the prodomain? We suggest cooperation between BDNFpro and mBDNF for convergent, but independent post-synaptic signaling. On the one hand, exogenous administration of mBDNF could restore LTP deficit in p75-flox mice[16], implicitly assuming that mBDNF supplied by astrocytes is sufficient for LTP maintenance. On the other hand, astrocytic expression of the sole BDNFpro in p75-flox slices (Fig. 5a) or its exogenous administration (Fig. 5c) equally rescued the LTP deficits. This suggests a mechanism, in which the basal levels of TrkB at spines cannot recruit sufficient mBDNF signaling for LTP maintenance. On recycling, BDNFpro increases TrkB levels at post-synaptic sites, allowing the receptor to act as a central controller. Accordingly, BDNF-TrkB signaling induces molecular adaptation underlining long-lasting synaptic strengthening[11] and shapes structural plasticity[54] via both TrkB tyrosine[55] and serine/threonine phosphorylation[56]. Given BDNFpro is not a ligand for TrkB[31] we do not expect a direct activation of any specific pathway initiated by the phosphorylation of selected tyrosines/serins in the catalytic domain of the receptor. However, providing that an increased and protracted TrkB expression is required for LTP maintenance, astrocytic BDNFpro release might indirectly be responsible for preferential activation of the specific phospholipase C-γ, PI3 kinase, and MAP kinase signaling that are required for LTP maintenance[23]. In support of this, we provided evidence that tyrosine phosphorylation at the phospholipase C-γ docking site of the TrkB receptor is increased by

the glial recycling process[16]. While this essential regulation saturates over time, the rescuing effect on LTP deficits by BDNFpro persists for the entire duration of the early-phase LTP expression, ending up at the initiation of the late-phase potentiation (Supplementary Fig. 7c). In marked contrast, mBDNF is only required for the first 10-20 min after stimulation, which is the predicted time of astrocytic proBDNF recycling[16]. Thus, prodomain and mature BDNF integrate various signaling pathways once LTP is expressed, and the conversion of the precursor acts as an inducible molecular switch. Notably, proBDNF can be also processed in the extracellular space by tissue plasminogen activator/plasmin[26], further modulating synaptic modification by the neurotrophin. Extracellular and astrocytic proBDNF processing are not conflicting mechanisms; they might reveal the molecular basis for two temporally distinct stages in LTP maintenance as reported previously[23]. While extracellular proBDNF processing by tPA provides cleavage of already existing proBDNF, astrocytic proBDNF processing focuses on newly synthesized proBDNF[14], which is recruited by peri-synaptic astrocytes for LTP maintenance. Additional mechanisms could control the extracellular availability of neurotrophin isoforms and the termination of their action. It has been shown that stimuli inducing hippocampal synaptic LTP enhance the motility of synapse-associated astrocytic processes[57,58]. This motility modifies the distance of microdomains from the stimulated synapses and is relatively rapid, which correlates with the approximate time of BDNFpro/mBDNF recycling in astrocytes. Structural remodeling of microdomains might be accompanied by changes in the ability of astrocytes to regulate proBDNF internalization, processing and recycling, and ultimately synaptic strengthening. In conclusion, proBDNF processing in astrocytes does not fully accomplish the classical model of self-sustained molecular alteration[1]; instead, it offers a switching mechanism from one inactive state to multi-functional one. This specifies strengthening mechanism for a persistent post-synaptic signaling, as it is required for a molecular memory to maintain LTP.

Spine targeting of TrkB tackles the underlying changes that occur at the synapse once LTP is expressed, reinforcing the common knowledge that LTP maintenance take place post-synaptically[6]. However, our data also provide evidence that the molecular basis responsible for these changes can be confined elsewhere. To understand the molecular foundation of LTP maintenance, it is therefore essential to know from which site of the synapse the molecular memory is originated. Since the discovery of LTP, most studies focused on whether pre- and/or post-synaptic sites are the compartments confining molecular modifications relevant for LTP maintenance; here we moved away from neurons and demonstrated that astrocytic microdomains are a central player for memory storage. Focus on the site of expression of BDNFpro was made possible by using electron microscopy (Fig. 3). BDNFpro-gold particles were seen at pre- and post-synaptic terminals and astrocytic microdomains closely interacting with these synaptic complexes. Gold particles residing in these synaptic sites account for highly intricate trafficking events induced by TBS: BDNFpro could be (i) synthesized at pre- and/or post-synaptic sites; (ii) transported to pre- and/or post-synaptic sites; (iii) internalized in pre- and/or post-synaptic compartments as well as in astrocytic microdomains. Given this complexity, it is of note that BDNFpro localization at astrocytic microdomains is the only relevant localization required for LTP maintenance. Although EM analysis showed a variety of synapses in terms of shapes and sizes, the one thing that was constant was the enrichment of BDNFpro-gold particle at peri-synaptic astrocytes (Fig. 3). This suggests the general occurrence of proteolytic conversion in this specialized area. Moreover, BDNFpro colocalized with the SNARE protein Vamp2 (Fig. 4a, b) and

intoxication with TeTN—a protease known to cleave Vamp2—inhibited BDNFpro secretion (Fig. 5b). This indicates that astrocytic vesicles store the processed neurotrophins at a place that is functional to synaptic use. Thus, astrocytic vesicles could serve as a reservoir of functional neurotrophin products that are available for synaptic needs. Such a modified and stable enrichment of astrocytic microdomains could serve as signaling hub of potentiated synapses providing an appealing model for information storage. The observation that persistent LTP and its associated synaptic changes are synapse-specific[59] raises the issue of how BDNF availability can be independently regulated in individual synapses. If BDNF could leak from a stimulated synapse, it would then massively, but incorrectly activate many additional synapses despite an appropriate stimulation. The idea of synaptic capture appears to be a valuable model in this context[59,60]. This implies that newly synthesized BDNF by neurons is delivered to all synapses, but it is only used at synapses that have been tagged by activity. In accordance with this model, TrkB, the correlated molecular tag for mBDNF[61], could capture mBDNF at selected synapses. Our finding that astrocytic BDNFpro increases TrkB/SorCS2 aggregation at the spine surface (Fig. 6c), supports the idea that proBDNF processing in microdomains may cooperate to selectively tag activated synapses. This specific process will then provide the capture of mBDNF for enhancing TrkB phosphorylation (Fig. 7c). Direct demonstration of the astrocytic origin of TrkB tagging would need to perform a two-pathway experiment[61] in perirhinal cortex of p75-flox mice; a brain area not permissive for this type of recording. However, this experiment conducted in hippocampus[43] confirmed that SorCS2 binds TrkB to facilitate TrkB/SorCS2 translocation at posts-synaptic sites for both synaptic tagging and LTP maintenance. Overall, our data suggest that astrocytic BDNFpro triggers mBDNF capture via TrkB tagging; thereby increasing selectivity and responsiveness to the neurotrophin at potentiated synapses.

The proposed critical role and mode of action of astrocytic proBDNF processing for LTP maintenance can be extended to long-lasting memory. Specifically, is proBDNF conversion important for memory consolidation? And as a consequence of this, is BDNFpro central to this type of memory? We demonstrated that viral expression of BDNFpro in perirhinal cortex of p75-flox mice resulted in mice that spent more time exploring a novel object than a familiar one during the 24-h-test phase (Fig. 8b). Astrocytic release of this byproduct is then correlated to long-lasting memory of the task. This finding supports the idea that astrocytic microdomains participate in changes in the strength of neuronal connections contributing to the most attractive cellular model for learning and memory, first defined by Hebb in 1949[62]. His hypothesis that synaptic strengthening for learning and memory occurs because of coincident activity between pre- and post-synaptic compartments now requires the integration of a new player, the astrocytic microdomains. This cyto-architectural structure accounts on functionally isolated subcellular domains that facilitate local homeostasis by redistributing ions, removing neurotransmitters, and releasing factors to influence moment-to-moment synaptic activity[8,63]. Our identification of an astrocytic molecular memory indicates that microdomains are also capable of sensing and integrating signals for persistent synaptic strengthening that are involved in memory consolidation in the brain. There is now overall agreement that persistent modification of the synaptic strength, via sustained LTP, represents a primary mechanism for the formation of memory engrams[64]. The contribution of astrocytes to synaptic engrams has been largely ignored in the past[9,65,66]; however, it is

evident that astrocytes appear to be an important cellular interface to control and modify neuronal data processing by their close physical contact with synapses[67]. Astrocytic microdomains could then provide local support over synaptic potentiation and the global control of neuronal ensembles engaged in memory circuits.

## Methods

**Experimental models**. All experiments were performed in accordance with European Union guidelines as approved by the institutional animal care and utilization committee (authorizations n°507/2017-PR; 76/2020 PR). p75-flox mice were generated by crossing loxP-p75$^{NTR}$-loxP mice kindly provided by B. Pierchala (University of Michigan School of Dentistry, USA) with GLAST-CreER$^{T2}$ Rosa-CAGloxP-stop-loxP(LSL)-R26R mice kindly provided by Prof. M Gotz (LMU, Munich, Germany). GFAP-GFP mice were kindly provided by Prof. A. Buffo (NICO, Torino, Italy). Animals were housed in a 12-h light/dark cycle with unrestricted access to food and water. For timed pregnancies, the plug date was designated as E0 and the date of birth was defined as P0.

**Tamoxifen treatment**. Male mice from p75-flox and control littermates were treated with 1 mg of tamoxifen (Sigma-Aldrich, Cat#T5648) dissolved in corn oil twice a day for 5 consecutive days at the age between P35 and P50.

**Stereotaxic surgery**. For virus delivery, 0 or 12 days after the last day of tamoxifen administration, mice were deeply anesthetized and viral particle (1 μl in volume) was infused into perirhinal cortex from each hemisphere (coordinates from Bregma: anteroposterior −2 mm, lateral ±4.2 mm, ventral +2.8 mm). Viral delivery was obtained through the insertion of capillary glasses (WPI) connected to a manual syringe pump (Narishige). Mice were allowed to recover and housed in standard cages until the day of sacrifice.

**Viral vectors**. pAAV.GFA104.PI.eGFP.WPRE.bGH (AAV-GFAP-GFP) adenoviral vector was used to express GFP in astrocytes specifically. This vector was a gift from Philip Haydon (Addgene viral prep #100896-AAV5; http://n2t.net/addgene:100896; RRID: Addgene_100896).

Lentiviral vectors were used for Cre-dependent expression of BDNFpro (LV-BDNFpro$^{stop}$), proBDNF (LV-proBDNF$^{stop}$), proBDNF$^{CR}$ (LV-proBDNF$^{CRstop}$) and light-chain of Tetanus toxin (TeTN) (LV-TeTN$^{stop}$). Lentiviral vectors were produced starting from pLV which drives gene expression under the cytomegalovirus (CMV) promoter. A gene-cassette including GFP encoding sequence was flanked by two modified loxP sites (lox2272). This prevented the formation of unwanted ATG start codons after Cre-mediated recombination. The GFP expression was stopped by two stop codons. Next, a single EcoRV-blunt cloning site, an IRES2 (internal ribosomal entry site2) followed by TandemTomato (Tomato) was cloned in 5′ direction before the WPRE lentiviral vector element. This construct showed strong GFP expression and weak expression of IRES2-Tomato. Under Cre-mediated recombination, the GFP was lost and the IRES2-Tomato cassette was under control of the promoter. Next, the EcoRV-blunt site was used to introduce the coding sequence of BDNFpro, proBDNF, proBDNF$^{CR}$, and light-chain of Tetanus toxin (TeTN). Lentiviral particles were produced in HEK293T cells. Lentiviral expression vectors were co-transfected with the pseudotyping vector pMD2.G and the packaging vector pCMVR8.91. Lentiviral particles were separated from the supernatant by ultracentrifugation and stored at −80 °C in 50 mM Tris-HCl, pH 7.4, 130 mM NaCl, 10 mM KCl, 5 mM MgCl₂.

To assess possible aversive effects of the viruses on the functional properties of astrocytes, we compared (i) BDNFpro expression in astrocytes from mice injected with AAV-GFAP-GFP with astrocytes from transgenic mice constitutively expressing GFP under the control of GFAP promoter (GFAP-GFP); (ii) LTP recording in mice injected with LV-BDNFpro$^{stop}$ and its control LV-GFP$^{stop}$ virus with non-injected wild-type mice; (iii) astrocytes morphology in mice injected with AAV-GFAP-GFP, LV-BDNFpro$^{stop}$ and its control LV-GFP$^{stop}$ which that of non-injected mice (iv) basic synaptic transmission (input/output) and synaptic facilitation (paired-pulse facilitation) in mice injected with LV-BDNFpro$^{stop}$ with non-injected mice.

**Electrophysiological recording**. Slices from perirhinal cortex were prepared from p75-flox, control littermates, or GFAP-GFP mice. The brain was removed and placed in cold oxygenated (95% O₂ and 5% CO₂) artificial cerebrospinal fluid (ACSF) containing 124.0 mM NaCl, 4.4 mM KCl, 1 mM NaH₂PO₄, 2.5 mM CaCl₂, 1.3 mM MgCl₂, 26.2 mM NaHCO₃, 10 mM glucose, and 2 mM L-ascorbic acid. Horizontal cortex slices (300-μm thick) were prepared using a vibratome and maintained in a chamber containing oxygenated ACSF at room temperature. After a minimum recovery period of 1 h, a single slice was transferred into a submersion recording chamber perfused (3 ml/min) with oxygenated ACSF at 32 °C ± 0.2 °C; square current pulses (duration 0.2 ms) were applied every 30 s (0.033 Hz) using a stimulus generator (WPI, stimulus isolator A360) connected through a stimulus isolation unit to a concentric bipolar electrode (40–80 KU, FHC) positioned in layers II/III on the temporal side of the rhinal sulcus. Evoked extracellular fEPSPs

were recorded using an Axoclamp-2B amplifier (Axon Instruments) with ACSF-filled glass micropipette pulled on a vertical puller (Narishige PC-10, resistance [<5 MU]), inserted in layers II/III at around 500 mm from the stimulation electrode, and analyzed using Axoscope 8.0 software. Baseline responses were obtained every 30 s with a stimulus intensity adjusted to induce 50% of the maximal synaptic response. After 20 min of stable baseline, LTP was evoked by TBS (100 Hz; four sets of stimulations delivered 15 s apart, each one consisting of ten bursts of five pulses at 100 Hz with inter-burst intervals of 150 ms). fEPSPs were plotted as amplitude. Each point represents the responses every 30 s expressed as means ± SEM.

Input/output curves were produced delivering stimuli (0.1 ms duration) to the stimulation electrode with stimulation intensities from 50 to 400 μA in steps of 50 μA.

Paired-pulse facilitation was expressed as the mean ratio of second and first fEPSP amplitude as a percentage with an interstimulus interval of 25, 50, 100, 150, and 200 ms.

In some experiments, recombinant BDNFpro (10 ng/ml; Alomone Labs, Cat#B-245), BDNFpro$^{Val/Met}$ (10 ng/ml; Alomone Labs, Cat#B-445), mBDNF (10 ng/ml; Laboratory of Antonino Cattaneo, SNS, Pisa, Italy), and proBDNF$^{CR}$ (20 ng/ml; Laboratory of Antonino Cattaneo, SNS, Pisa, Italy) were perfused into a recording chamber.

In some experiments, slices were preincubated with plasmin (100 nM) for 2 h and were maintained in perfusion for the whole recording time.

**Immunohistochemistry**. Brain slices were fixed in 4% PFA for 1 h after recording. Slices were treated with 1% Triton X-100 for 20 min, blocked with 3% BSA in PBS for 1 h, and incubated overnight free-floating with primary antibodies diluted in blocking buffer. Slices were washed in PBS and incubated for 2 h at room temperature with secondary antibodies diluted in blocking buffer. Slices were eventually counterstained with DAPI (Sigma-Aldrich, Cat#D9542) and mounted with Aqua Poly/mount (Polysciences, Inc., Cat #18606).

**Antibodies**. The following antibodies were used: rabbit α-GFP (Thermo Fisher Scientific Cat#A-6455; RRID:AB2536208; IHC 1:1000 ICC 1:1000), chicken α-GFP (Thermo Fisher Scientific Cat#A10262; RRID:AB2534023, IHC 1:1000), chicken α-BDNF (Promega Cat#G1641; RRID:AB#430850, IHC 1:300; WB 1:500), rabbit α-BDNF (Alomone Labs Cat# ANT-010; RRID:AB_2039756, EM 1:20), chicken α-proBDNF (Millipore Cat#AB9042; RID:AB2274709, IHC 1:300), rabbit α-proBDNF (Alomone Labs Cat#ANT-006; RRID:AB_2039758, EM 1:20), rabbit α-BDNFpro (Laboratory of Bai Lu, Govern Institute for Brain Research, Tsinghua University, Beijing, IHC 1:300, EM 1:20, WB 1:500), mouse α-NeuN (Abcam Cat#ab77315; RRID:AB#1566475, IHC 1:1000), guinea pig α-NeuN (Millipore Cat#ABN90; RRID:AB#11205592, IHC 1:1000), rabbit α-pTrkB (Tyr 816) (Laboratory of Moses Chao, Skirball Institute of Biomolecular Medicine, New York, USA, IHC 1.25 mg/ml), goat α-TrkB (Santa Cruz Biotechnology Cat#sc-12-G; RRID:AB#632558, IHC 1:300), mouse α-PSD95 (Merck-Millipore Cat#MAB1596; RRID:AB_2092365, IHC 1:500), sheep α-SorCS2 (R&D System Cat#AF4238; RRID:AB_10645642), rabbit α-SorCS2 (MyBioSource Cat# MBS5302436), rabbit α-p75$^{NTR}$ (Promega Cat#G3231; RRID:AB_430853, IHC 1:1000), chicken α-GFAP (Abcam Cat#ab134436; RRID:AB_2818977, IHC 1:1000), mouse α-GFAP (Abcam Cat#ab10062; RRID:AB_296804, IHC 1:1000), mouse α-synaptobrevin2\VAMP2 (Synaptic System Cat#104 211; RRID:AB_2619758, IHC 1:1000), rabbit α-beta galactosidase (Proteintech Cat#15518-1-AP; RRID:AB_2263448, IHC 1:500), rabbit α-RFP (Rockland Antibodies Cat#600-401-379; RRID:AB_2209751 IHC 1:1000), mouse α DsRed (Santa Cruz Biotechnology Cat# sc-390909; RRID:AB_2801575 IHC 1:1000).

**Western blot experiments**. Recombinant mBDNF, proBDNF$^{CR,}$ and BDNFpro were suspended in protein sample buffer, boiled for 7 min at 95 °C, resolved by 12% SDS-PAGE, and transferred to nitrocellulose membranes. Membranes were incubated (1 h) at room temperature in blocking solution (5% BSA in TBS-Tween 0.1%) and incubated overnight at 4 °C with primary antibodies. Membranes were washed three times in TBS-Tween and incubated for 1.5 h at room temperature with horseradish peroxidase (HRP)-conjugated secondary antibodies. ECL was used for detection.

**Primary cell cultures**. Cortices were isolated from E17.5 mouse embryos and incubated for 20 min at 37 °C in Trypsin/EDTA 0.25% (Thermo Fisher Scientific, Cat#25200056). Cortical cells were dissociated with a plastic pipette, collected by centrifugation (800 RPM, 5 min), and resuspended in Dulbecco's modified Eagle's medium (DMEM; GIBCO Cat#10938025) supplemented with 10% of fetal bovine serum and Penicillin/Streptomycin (GIBCO; Cat#15070063). Cells were transfected using Amaxa Nucleofector system following manufacturer instructions and plated on glass coverslips pre-coated with poly-L-lysine (0.1 mg/ml; Sigma-Aldrich, Cat#P6282). Three hours after plating, the medium was changed to Neurobasal (GIBCO, Cat#21103049) supplemented with B27-supplement (GIBCO, Cat#17504044) and Penicillin/Streptomycin/Glutamine (GIBCO, Cat#1038016).

**Proximity ligation assay (PLA)**. PLA was performed on fixed mouse primary cortical neurons non-permeabilized and on fixed perirhinal sections permeabilized

with Triton X 1% for 20 min. PLA was performed following the manufacturer's instruction (Duolink PLA Sigma-Aldrich). Briefly, samples were blocked with 3% BSA in PBS for 1 h and incubated with primary antibodies at 4 °C overnight. Samples were washed with PBS and incubated with PLA probes MINUS (Duolink PLA Probe Anti-Goat MINUS antibody, Sigma-Aldrich Cat# DUO92006) and PLUS (Duolink PLA Probe Anti-Rabbit PLUS antibody, Sigma-Aldrich Cat# DUO92002) for 1 h at 37 °C. The PLA probes were diluted (1:5) in antibody diluent. Samples were washed in 1X buffer A 2 × 5 min under gentle agitation and incubated with ligation-ligase solution for 30 min at 37 °C. Samples were washed in 1X wash buffer for 2 × 2 min under gentle agitation and incubated with amplification polymerase solution for 90 min at 37 °C. In some experiments, we performed immunodetection over PLA incubating PLA samples with secondary antibodies coupled with the appropriate fluorophores for 2 h at room temperature. Samples were washed in 1X wash buffer B for 2 × 10 min and with 0.01X wash buffer B for 1 min. Samples were mounted with Prolong Gold antifade reagent with DAPI.

**Confocal microscopy and quantitative image analysis**. Confocal imaging was performed using a laser-scanning motorized confocal system (Nikon A1) equipped with an Eclipse Ti-E inverted microscope and four laser lines (405, 488, 561, and 638 nm). Z-series images were taken with an inter-stack interval of 0.5 μm using ×60 oil objective. For each type of quantification, laser intensities and camera settings were maintained identically within the same experiment to allow the comparison of different experimental groups and treatments. Image processing and 3D-rendering were performed using the software NIS Element (Nikon). Colocalizations (BDNFpro/GFP; Vamp2-GFP; BDNFpro/Vamp2; TrkB/pTrkB; PLA$^{TrkB/}$$^{SorCS2}$/NeuN; PLA$^{TrkB/SorCS2}$/PSD95) were quantified calculating Mander's overlap or Pearson's correlation using the software NIS Element (Nikon). Colocalization data for the comparison of stimulated slices (TBS) vs. non-stimulated slices (baseline) are expressed following normalization of the Mander's overlap (co-occurrence) and Pearson's correlation (correlation) of each treatment to the average of the baseline. In the specific context of our work, co-occurrence was preferred to correlation to measure co-localization between the BDNFpro and the cellular filler GFP, which define the astrocytic territory. We adopted co-occurrence as it satisfied the following requirements (Aaron et al.[24]): (i) it accounts for the abundance of fluorophores that overlap with each other; (ii) it gives greater importance to pixels that are intense and minor relevance to those which intensity is near the threshold; (iii) it ameliorates the effects of unwanted signals, such as auto-fluorescence or other near-threshold signals; (iv) it can offer an intuitive accounting of the concentrated weighted overlap between two imaging targets, with relative insensitivity to imaging noise; (v) it shows great ability to assess to what extent a structure or molecule can be found in a particular location in an intensity-weighted manner. These features acquire clear relevance for the present experiments which require the use of thick slices (300 μm) for the preservation of neuronal and glial network. While maintaining deep functional synaptic contacts, they are challenging for antibody penetration and deep confocal or SIM detection increasing the relevance of auto-fluorescence or other near-threshold unwanted signals.

**3D-SIM**. 3D-SIM was performed using X-Light V2 confocal spinning disk system completed with a Video Confocal super-resolution module (CrestOptics, Italy) with a lateral resolution of 115 nm and an axial resolution of ~250 nm. The system was equipped with a 60x/1.40 NA PlanApo Lambda oil immersion objective (Nikon, Japan), Zyla sCMOS camera (Andor), and Spectra X Lumencor LEDs Light Source with bandpass excitation filters of 460–490 and 535–600 nm (Chroma Technology, USA). Image stacks were acquired with a format of 2048 × 2048 pixels and a z-distance of 150 nm and with 36 raw SIM images per plane (multiple acquisition mode x–y grid scan). The SIM raw data with 16-bit depth were computationally reconstructed using the Metamorph software package. For 3D-image rendering and colocalization analysis, images were processed using NIS-Elements software. Quantification of BDNFpro/GFP puncta was performed manually counting the number of immunoreactive puncta in finger-like extension and lamellar sheaths compared to the total number of puncta found in astrocytes (for these experiments n = 6 cells, 3 slices, 3 mice for each condition).

**Electron microscopy**. Sections (300 μm) from electrophysiological recordings were fixed in a solution of 4% PFA and 0.1% glutaraldehyde in 0.1 M phosphate buffer (PB) for 4 h and then washed overnight in 50 mM TBS, pH 7.4. Slices were immersed for 2 h in a cryoprotecting solution (25% sucrose and 10% glycerol in 0.05 M PB) before being snap-frozen in liquid nitrogen-cooled isopentane. After washing in 0.1 M PB, sections were incubated for 1 h in 20% normal goat serum in TBS. Slices were then incubated for three nights at 4 °C in a 3% BSA/TBS solution containing primary αBDNFpro or αmBDNF or αproBDNF. Sections were incubated overnight at 4 °C in 3% BSA/TBS solution containing 2-nm-gold conjugated goat arabbit (BBI International). Sections were rinsed in TBS and then fixed for 30 min in a 2.5% glutaraldehyde solution and post-fixed with 1% osmium tetroxide. After dehydration in an ethanol series followed by propylene oxide, samples were embedded in Epon812 epoxy resin and observed with a JEOLJEM-1011

transmission electron microscope, operated at 100 kV, after cutting ultrathin sections (60-nm thick). Negative controls were performed in the absence of primary antibody.

Astrocytes were identified using 3 structural criteria: (i) localization between synapses: the space between synapses is occupied by astrocytic processes. Astrocytes are indeed cellular structures just filling all the space between synapses, dendrites, and axons; (ii) irregular, stellate shape: the processes of astrocytes surrounding synaptic contacts often adopt quite convoluted forms. In comparison to their flatter neuronal counterparts, perisynaptic astrocytes show an irregular profile; (iii) relatively clear cytoplasm: the cytoplasm of an astrocyte differs from that of neighboring objects in the neuropil. An astrocyte cytoplasm is less electron-dense than the one of neurons. Quantification of gold grains was performed manually counting the number of gold grains (830 grains; 41 sections; 5 slices; 3 mice) labeling astrocytic microdomains that surround pre- and post-synaptic structures. Moreover, gold grains were counted in a membrane delimited area of 230 nm radius surrounding synaptic contacts compared to the total number of gold grains found in astrocytes.

**Object recognition test (ORT)**. ORT was performed in a Y-apparatus with high, homogenous white walls, 30-cm high: one arm was used as start arm and had a sliding door to control access to the arena; the other two arms were used to display the objects. The start arm is 26 cm in length with the sliding door placed at 13 cm, and the lateral arms are 18-cm long. All arms are 10-cm wide. On the first day (habituation day) mice explored the arena for 20 min. The following day, the learning session (sample phase) was performed: for 5 min the animals were free to explore the arena, which contained two identical objects, one for each arm, placed at the end of the arm. At 10 min or 24 h after the sample phase, the test phase was run; the animals were free to explore the arena containing two objects, the familiar object and a novel one. Arena and objects were cleaned up between trials to stop the build-up of olfactory cues. The discrimination index was calculated as follows: (TNew − TOld)/(TNew + TOld), with TNew and TOld being the time spent exploring the new and the familiar objects, respectively. A video camera was mounted above the apparatus to record trials with the EthoVision software (Noldus). The exploration time was taken as the time during which mice approached the objects with muzzle and paws.

**Statistics and reproducibility**. All data were run for a normality test before a statistical comparison test. Data normally distributed were summarized by mean ± SEM. Comparison between two different groups was assessed by an unpaired t-test. For ORT discrimination index, two-way ANOVA followed by a Holm–Sidak method test was used. The level of significance used was $p < 0.05$.

**Reporting summary**. Further information on research design is available in the Nature Research Reporting Summary linked to this article.

## Data availability
The source data underlying Figs. 1, 3, and 5–8 are provided as Supplementary Data 1. The original uncropped blots referring to Fig. 1a are provided in Supplementary Fig. 1. Any other relevant data are available upon request.

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

## Acknowledgements

We thank Massimo Gamberini and Silvia Biggi for technical assistance and Ranzi Barbara for lab management; Moses Chao for providing pTrkB antibody; Brian A. Pierchala for providing p75lox/lox mice; Antonino Cattaneo for providing recombinant BDNF and proBDNF^CR; Magdalena Götz for providing GLAST-CreER^T2 mice; Annalisa Buffo for providing GFAP-GFP mice; IN-BDNF consortium for scientific support. This project received funding from Progetti di Ricerca di Rilevanza Nazionale (PRIN), Bando 2017 Project number 2017HPTFFCPRIN to M.C.; European Research Council (ERC) under the European Union's Horizon 2020 research and innovation program (grant agreement N° 788793-BACKUP) to M.C. and B.V.; Deutsche Forschungsgemeinschaft project-ID 194101929-BL567/3-2 and project ID44541416–TRR58–A10 to R.B.; Graduate School of Life Sciences (GSLS) Würzburg fellowship to M.S.; Fondazione Umberto Veronesi post-doctoral grant 2018 to B.V; Fondazione Umberto Veronesi to G.S.

## Author contributions

Conceptualization: B.V. and M.C.; investigation: B.V., S.S., M.S., V.B., and G.S.; writing—original draft: B.V. and M.C.; writing—review and editing: R.B. and B.L.; supervision: R.B., R.R.G., S.S., E.B., and N.B.; funding acquisition: M.C., B.V., and R.B.; resources: B.L., R.B., and S.S.

## Competing interests

The authors declare no competing interests.
