## [Transparent Peer Review File · Communications Biology]

Reviewers' comments:

Reviewer #1 (Remarks to the Author):

In the manuscript by Beatrice Vignoli and co-workers, authors hypothesis that astrocytes not only can internalize proBDNF released by neurons but also are able to convert it into active prodomain (BDNFpro) and mature BDNF (mBDNF). In addition, they describe a novel function of astrocytic BDNFpro as an enhancer of post-synaptic TrkB signaling.

To verify first hypothesis, authors applied a couple of state-of-the art imaging techniques (e.g. SIM and EM) in combination with functional electrophysiological analysis using acute mouse brain slices as a model. Using an antibody specifically recognizing furin cleavage site, they shown accumulation of BDNFpro within the territory covered by astrocytes following LTP-inducing electrical stimulation (TBS). Moreover, 3D-SIM imaging as well as EM analysis provided evidence for vesicular localization of BDNFpro in astrocytic microdomains.

To verify the second hypothesis, authors combined PLA assay with electrophysiological measurements in slices of genetically modified mice and behavioral approaches.

This is very interesting, topical and methodically sophisticated study demonstrating an additional facet about the role of astrocytes in synaptic plasticity. In particular, the second part of this study provide clear evidence for the novel role of astrocytic BDNFpro for TrkB/SorCS2 aggregation/signaling.

However, some data are not entirely convincing, several points require further clarification and, likely, additional experiments must be performed (s. below) before this manuscript can be accepted for publication.

Major concerns:

(1) To my opinion, current experimental results do not support the idea for proteolytic conversion of proBDNF into mBDNF and BDNFpro within the astrocytic microdomains. Although authors clearly demonstrated presence of BDNFpro in astrocytic vesicles and its increase after TBS, direct evidence for astrocytic cleavage is still missing. One possible source of BDNFpro in astrocytes might be e.g. its uptake from the extracellular space.

It has been shown that proBDNF undergoes extracellular cleavage by tPA/plasmin during LTP induction, while mBDNF at maintenance stage is derived from intracellular cleavage of proBDNF by furin/PC1. I suggest that authors could use the knowledge about BDNF processing to perform experiments using a membrane impermeable inhibitor for tPA and a membrane permeable inhibitor for FIN/PC1 followed by biochemical, imaging and functional analysis to directly demonstrated existence of intracellular cleavage of proBDNF within astrocytic microdomains.

(2) LTP induction involves BDNF-TrkB signaling mainly through MAP kinase, whereas phospholipase C- γ , PI3 kinase, and MAP kinase are required for the LTP maintenance. Which signaling pathways become activated via BDNFpro?

(3) Figs. 1 and 4a-b show co-localization of BDNFpro with astrocytic territory. However, because of insufficient resolution, these results can be contaminated by BDNFpro resided within neurons. To exclude this possibility, I suggest to perform similar experiments after expression of additional fluorescent protein (e.g. mCherry or tdTomato) under control of neuronal promoters followed by separate quantitative co-localization analysis for astrocytes and neurons.

(4) Fig. 2. Please add synaptic staining and perform quantitative co-occurrence analysis separately for GFP/BDNFpro and synaptic marker/BDNFpro (s. also comment 2 above).

(5) What is the absolute amount of BDNFpro-positive gold particles? Is it correspond to results shown in Fig. 1 (ctrl and TBS)?

Minor concerns:

(1) Whether the amount of proBDNF/mBDNF within astrocytic territory (as shown in Fig. 1) also changes in control versus TBS-treated slices? Authors could use mBDNF antibody (as shown in Supp. Fig. S3) to analyze it.

(2) Fig. 1a. Authors' should provide Western blot showing expression of the housekeeping protein (e.g. GAPDH) for all lines. They also should provide Western blot for the same samples probed with mBDNF antibody.

(3) In Supp. Fig. S1d, authors use Pearson's coefficient to analyze correlation between BDNFpro/GFP, while in all other figures they applied Mander's coefficient to analyze co-occurrence. What is the reason for the different analysis?

Reviewer #2 (Remarks to the Author):

In this article, Vignoli et al. cleverly used viral strategies to show step by step how the BDNFpro of the astrocytic compartment was docked with Vamp2 secretory vesicles and could rescue LTP. Astrocytes thus appear as proficient for BDNFpro release to allow the late-phase LTP by increasing TrkB/SorCS2 aggregation and TrkB phosphorylation and allow memory retention. In addition, the authors investigated whether the Val66Met mutation, known to induce memory alteration in humans, could alter the bioactivity to BDNFpro.

It is a very well-constructed and clearly written article. This study sheds new light on the understanding of the role of astrocytic BDNFpro and microdomains for the confinement of a "molecular memory". This work represents an impressive number of experiments and makes a fundamental contribution to our understanding the role of astrocytes in learning and memorization.

I only have 2 major questions:

1. In this study, the authors used a large number of different lenti and AAV viruses that can induce astrocyte reactivity and thus the release of gliotransmitters. Do the authors use controls to assess astrocytic reactivity that could interfere with the interpretation of their results?

2. As there are also TrkB receptors located on astrocytes, could a mBDNF feedback loop on astrocytes also play a role in the observed mechanism? Could you test the invalidation of the TrkB receptors in astrocytes to assess its impact on LTP induction and maintenance? As astrocytic processes can regulate their distances to synapses and thus contribute to the regulation of synaptic transmission, the proximity of astrocytic processes to synapses could also be modulated following your stimulation of TBS. This could be discussed as well.

And few minor questions:

3. Figure 1 appears to come from the same experiment also illustrated in Figure 1 of Vignoli et al. 2016. Yet, the % co-localization induced by the TBS is much lower. If it is the same experiment, why are these values different? If not, why did the authors change the conditions that would have been more advantageous?

4. In the supplementary figure 1, middle and right panels are supposed to be from the same ROI.

However, there is more signal on the right panel, that is logically not possible?

5. In Figure 2b ROI n °4, the BDNF pro/GFP signal is not visible. Another illustration should be found.

6. In the supplementary figure 2c, the histogram should be replaced by individual dot plots. Idem for Fig. 6c and all histograms.

7. No statistics were made to compare the late phases of LTP. Although the effects are clear, statistics should be added.

8. In figure 3c, the dashed square for the zoom n °5 is not correctly placed. Quantifications could also be added to the figure in order to easily access the variability and the quantity of quantified dots.

9. P13 line 330 please reformulate " ... that indicates that".

10. P14 line 347: I did not understand why the authors measured only pTrkB that colocalized with TrkB and not the total immunoreactivity of neuronal pTrkB. Please clarify.

11. In the discussion section, the signs ==> should be replaced by (i) and (ii), line 405.

Point-by-point revision

Reviewer #1

We thank the Reviewer#1 for the overall appreciation of our findings. Concerning Her/His remarks:

Point 1. To my opinion, current experimental results do not support the idea for proteolytic conversion of proBDNF into mBDNF and BDNFpro within the astrocytic microdomains. Although authors clearly demonstrated presence of BDNFpro in astrocytic vesicles and its increase after TBS, direct evidence for astrocytic cleavage is still missing. One possible source of BDNFpro in astrocytes might be e.g., its uptake from the extracellular space.

We agree that a direct evidence for proBDNF cleavage in astrocytes is missing; contextually we have now smoothed the text emphasizing that our findings only “suggest” for astrocytic proBDNF processing. Direct biochemical demonstration of proBDNF cleavage will require the use of cultured astrocytes, which stand for a simplified experimental setting that won't maintain the complexity of proBDNF trafficking described in this work. Astrocyte cultures cannot consider (i) proBDNF secretion from neurons following TBS; (ii) proBDNF internalization in microdomains; (iii) proBDNF proteolytic processing; (iv) secretion of the proteolytic products BDNFpro and mBDNF. Thus, our demonstration of proBDNF processing was necessarily indirect and focused on the specific detection of BDNFpro by using an antibody for the furin cleaved C-terminal end of the prodomain, together with the use of p75-flox mice, which prevent from proBDNF internalization and recycling in astrocytes following TBS (Vignoli et al., 2016).

Despite these constrains we have now included additional experiments. We demonstrated that the astrocytic uptake of proBDNF was abolished upon prior incubation with *plasmin*, an enzyme responsible for the extracellular proteolysis of proBDNF to pro-domain and mature protein. Depletion of proBDNF and concomitant increase of BDNFpro and mBDNF proteolytic products in the extracellular space induced by the enzyme prevented from the expression of every neurotrophin isoforms in astrocytes (Fig. 1e). This new experiment clearly implies that neither BDNFpro nor mBDNF proteolytic products show individual internalization in astrocytes; moreover, it implies that major source of the proteolytic products in these cells likely comes from intracellular processing of previously internalized proBDNF. Accordingly, the uptake of proBDNF in astrocytes was prevented by specific deletion of its carrier receptor p75^{NTR} (Vignoli et al., 2016). Given BDNFpro shows no binding to p75^{NTR} (Anastasia et al., 2013), the lack of BDNFpro expression in p75^{NTR}-deficient

astrocytes (Fig. 1f) implies that the uptake of this isoform from the extracellular space is unlikely to occur.

It has been shown that proBDNF undergoes extracellular cleavage by tPA/plasmin during LTP induction, while mBDNF at maintenance stage is derived from intracellular cleavage of proBDNF by furin/PC1. I suggest that authors could use the knowledge about BDNF processing to perform experiments using a membrane impermeable inhibitor for tPA and a membrane permeable inhibitor for FIN/PC1 followed by biochemical, imaging and functional analysis to directly demonstrated existence of intracellular cleavage of proBDNF within astrocytic microdomains.

One way to discriminate among extracellular and intracellular proBDNF cleavage is to eliminate one process at a time. The reviewer suggested to use:

- (i) “a membrane impermeable inhibitor for tPA”. By inhibiting tPA activity, the extracellular proBDNF will remain uncleaved and presumably available for endocytosis in astrocytes. Accordingly, the presence of BDNF_{pro} immunoreactivity in these cells should stand for an “intracellular cleavage of proBDNF within astrocytic microdomains”. However, one should take into account the intrinsic limits of this pharmacological approach. Additional tPA-dependent mechanisms as e.g., tPA endocytosis and recycling by astrocytes could also be involved (Cassé et al., 2012). The presence of a tPA recycling pathway in astrocytes will strongly affects the interpretation of the results. To overcome this limitation, we have used *plasmin* and directly degraded the extracellular pro-neurotrophin (see above).
- (ii) “a membrane permeable inhibitor for FIN/PC1”. To demonstrate the “existence of intracellular cleavage of proBDNF” in astrocytes the enzymatic inhibition of *FIN/PC1* should selectively target the pro-neurotrophin in these cells. However, such a compound does not distinguish between different cellular types, e.g., neurons from astrocytes or other possible cellular players (i.e., microglia). Given the endocytic proBDNF in astrocytes is sourced by neurons, it is essential to discriminate proBDNF processing at least between these two different cellular types. Moreover, *furin* is a general proteolytic enzyme that provide maturation and processing of many different proteins in different cells, including those that are relevant for synaptic strengthening. Broad inhibition of *furin* will strongly prevent the interpretation of the results.

To this point, we would like to emphasize that extracellular and astrocytic proBDNF processing are not conflicting mechanisms. These mechanisms might reveal the molecular basis for two temporally distinct stages in LTP maintenance as reported by Pang et al., (2016). While extracellular proBDNF processing by tPA provides cleavage of already existing proBDNF, astrocytic proBDNF processing focuses on newly synthesized proBDNF, which excess is recruited by peri-synaptic astrocytes for LTP maintenance. This issue has now been discussed in the revised version of our manuscript.

Point 2. LTP induction involves BDNF-TrkB signaling mainly through MAP kinase, whereas phospholipase C- γ , PI3 kinase, and MAP kinase are required for the LTP maintenance. Which signaling pathways become activated via BDNFpro?

Our major finding is that astrocytic BDNFpro increases the surface expression of TrkB at post-synaptic sites, leaving the receptor to be activated by mBDNF according to synaptic need. Given BDNFpro is not a ligand for TrkB (Anastasia et al., 2013) we do not expect a direct activation of any specific pathway initiated by the phosphorylation of selected tyrosines/serins in the catalytic domain of the receptor. However, providing that an increased and protracted TrkB expression is required for LTP maintenance, we favor that glial BDNFpro release is indirectly responsible for preferential activation of the specific “*phospholipase C- γ , PI3 kinase, and MAP kinase*” signaling as mentioned by the Reviewer#1. In line of this, we provided evidence that tyrosine phosphorylation at the phospholipase-C γ 1 docking site of the TrkB receptor is increased by the glial recycling process (Fig.7b). This issue has now been discussed in the revised version of our manuscript.

Point 3. Figs. 1 and 4a-b show co-localization of BDNFpro with astrocytic territory. However, because of insufficient resolution, these results can be contaminated by BDNFpro resided within neurons. To exclude this possibility, I suggest to perform similar experiments after expression of additional fluorescent protein (e.g. mCherry or tdTomato) under control of neuronal promoters followed by separate quantitative co-localization analysis for astrocytes and neurons.

One way to discriminate among synaptic and astrocytic BDNFpro is to eliminate one of the two component and quantify what is left of the other one. Control experiments in this sense were presented in our original manuscript. When BDNFpro internalization was prevented in p75^{NTR}-deficient astrocytes BDNFpro/GFP co-localization was reduced to background levels (Fig. 1f). Thus, the contribution of synaptic BDNFpro to the overall amount of BDNFpro in the astrocytic territory is minute, if not absent, and did not affect the relevance of our quantification analysis. This control experiment also applies to Point 4.

Unfortunately, this aspect was not clearly presented in our original manuscript; to avoid any misinterpretation by the reader we have now emphasized this important control in the revised version of our manuscript.

Point 4. *Fig. 2. Please add synaptic staining and perform quantitative co-occurrence analysis separately for GFP/BDNFpro and synaptic marker/BDNFpro (s. also comment 2 above).*

See *Point 3*.

Point 5. *What is the absolute amount of BDNFpro-positive gold particles?*

We performed *pre-embedding* labeling of proBDNF, BDNFpro and mBDNF. This procedure involves a labeling step before the embedding of the specimen in the resin, its subsequent sectioning and the analysis of sections under the electron microscope. Labeling with 2 nm gold particles is required to partially overcome the impairment of antibody penetration in pre-embedded tissues. The use of larger (5 and 10 nm) gold particles, failed any tissue penetration of the antibodies (Dieni et al., 2008; Vignoli et al., 2016).

While these are non-permissive conditions for an absolute BDNFpro-gold quantification, a relative estimation (percentage of the total number of gold particles detected in astrocytes) of BDNFpro-gold at peri-synaptic astrocytes was originally reported in the text and now showed in Fig. 3d. To easily access our quantification analysis, we added a new panel in which we quantified the distribution of BDNFpro-gold (per section) in whole astrocytes and peri-synaptic astrocytes. Moreover, a detailed information about total number of (i) gold particles; (ii) EM sections; (iii) cortical slices and (iv) mice is now reported in the Methods section.

Is it correspond to results shown in Fig. 1 (ctrl and TBS)?

In Fig. 1 and 3 we showed BDNFpro immunoreactivity using different detection methods. While Fig. 1 displays BDNFpro immunofluorescence in the whole astrocytic territory, Fig. 3 restrict the localization of BDNFpro immunogold at single peri-synaptic sites. A correspondence of these analysis would require correlating light and electron microscopy technique, thereby to localize BDNFpro at the ultrastructural level in the same biological tissue. However important limitations still exist with regard to how well different types of imaging information can be correlated, particularly for quantification analysis. These limitations are primarily due to the (i) different resolution of the two microscopy techniques (confocal vs. EM); (ii) different detection methods (immunofluorescence vs. immunogold); and (iii) sample preparation constraints. Contextually we tried a *post-embedding* protocol, in

which tissues were embedded in a hydrophobic resin and ultrathin sections are first obtained and then immunolabeled. These conditions are compatible to immunohistochemistry; however, as also confirmed in Dieni et al., (2012), most antigenicity to our molecules of interest was lost. Thus, a direct comparison between the extent of BDNFpro-gold with that of BDNFpro immunofluorescence was impractical.

Point 6. *Whether the amount of proBDNF/mBDNF within astrocytic territory (as shown in Fig. 1) also changes in control versus TBS-treated slices? Authors could use mBDNF antibody (as shown in Supp. Fig. S3) to analyze it.*

We now added the new requested experiments in Fig. 1d of our revised manuscript.

Point 7. *Fig. 1a. Authors' should provide Western blot showing expression of the housekeeping protein (e.g. GAPDH) for all lines.*

Our Western Blot is made by loading defined concentrations of recombinant proteins (cleavage-resistant proBDNF^{CR}, mBDNF and BDNFpro) simply diluted in loading buffer. It does not contain lysates from cells or tissues and, therefore, no "housekeeping protein (e.g. GAPDH)" are present for detection.

They also should provide Western blot for the same samples probed with mBDNF antibody.

We have now included the requested Western blot in Fig. 1a of our revised manuscript.

Point 8. *In Supp. Fig. S1d, authors use Pearson's coefficient to analyze correlation between BDNFpro/GFP, while in all other figures they applied Mander's coefficient to analyze co-occurrence. What is the reason for the different analysis?*

While Mander's coefficient describes the extent of spatial overlap between two fluorophores (*co-occurrence*), the Pearson's correlation determines the degree to which the abundance of two spatially overlapping fluorophores are related to each other (*correlation*). In the specific context of our work, *co-occurrence* is ideal to measure colocalization between the BDNFpro and the cellular filler GFP, which define the astrocytic territory (spatial overlap). However, both measures (*co-occurrence* and *correlation*) can supply strengthening information about our biological system, and for this reason we also reported one representative example of *correlation* analysis. By using Pearson's *correlation*, we confirmed that BDNFpro/GFP colocalization in astrocytes is increased upon TBS.

We adopted co-occurrence as it satisfied the following requirements (from Aaron et al., 2018):

- (i) it accounts for the abundance of fluorophores that overlap with each other;
- (ii) it gives greater importance to pixels that are intense and minor relevance to

- those which intensity is near the threshold;
- (iii) it ameliorates the effects of unwanted signals, such as auto-fluorescence or other near-threshold signals;
 - (iv) it can offer an intuitive accounting of the concentrated weighted overlap between two imaging targets, with relative insensitivity to imaging noise;
 - (v) it shows great ability to assess to what extent a structure or molecule can be found in a particular location in an intensity-weighted manner.

These features acquire clear relevance for the present experiments which require the use of thick slices (300 μm) for the preservation of neuronal and glial network. While maintaining deep functional synaptic contacts, they are challenging for antibody penetration and deep confocal or SIM detection increasing the relevance of auto-fluorescence or other near-threshold unwanted signals. In the revised version of our manuscript, we have now detailed why we adopted Mander's coefficient.

Reviewer #2

We thank the Reviewer #2 for Her/His enthusiastic appreciation of our work. Concerning the reviewer's remarks:

Point 1. In this study, the authors used a large number of different lenti and AAV viruses that can induce astrocyte reactivity and thus the release of gliotransmitters. Do the authors use controls to assess astrocytic reactivity that could interfere with the interpretation of their results?

To eliminate doubts that the viral transduction could possibly alter the functional properties of astrocytes,

- (i) we repeated most of our experiments in which we used different viruses, utilizing transgenic mice constitutively expressing GFP under the control of GFAP promoter (GFAP-GFP) (Supplementary Fig.1a and b).
- (ii) we analyzed whether in our experimental conditions astrocytes predispose to cytoskeleton hypertrophy, a feature of astrogliosis, and showed their morphology (Fig. 1; Fig. 2; Fig. 4; Supplementary Fig. 5).
- (iii) we assessed the possibility that in our experimental condition astrocytes may differ in their physiology (e.g., gliotransmission) even if they exhibit normal morphology. We performed experiments in which we controlled various aspects of synaptic activity upon viral transduction. We demonstrated that both basic synaptic transmission (input/output) and synaptic facilitation (paired-pulse facilitation) were comparable in viral transduced and non-transduced slices (Supplementary Fig. 5).

Unfortunately, these aspects were not clearly presented to the readers; to avoid any misinterpretation of our data we have now explained them better in the Methods section of our revised manuscript.

Point 2. As there are also TrkB receptors located on astrocytes, could a mBDNF feedback loop on astrocytes also play a role in the observed mechanism?

The expression of neurotrophin receptors by astrocytes is still a matter of debate. Conflicting data from several publications confirmed and/or excluded the presence of neurotrophin receptors in these cells (Brigadski and Leßmann, 2020), encouraging to test them in specific experimental conditions. In our experimental context, we reported that astrocytes from perirhinal cortex express p75^{NTR} and TrkB-truncated but not full-length TrkB (Bergami et al., 2008). Moreover, p75^{NTR}, but not TrkB-truncated, was internalized upon treatment with

proBDNF (Bergami et al., 2008), indicating that neurotrophin internalization can only occur after binding to p75^{NTR}.

Could you test the invalidation of the TrkB receptors in astrocytes to assess its impact on LTP induction and maintenance?

To readily assess the role of the astrocytic TrkB in LTP, we used TrkB^{ff} mice (Minichiello et al., 1999) injected with AAV-Cre virus under the control of GFAP promoter for selective deletion of *TrkB-gene* in astrocytes. LTP was induced applying θ -burst-stimulation in layer II/III of perirhinal cortices. In slices from *TrkB-gene* deleted mice TBS induced an increase in fEPSP amplitude that remained potentiated above baseline for the 3h duration of the recording. In agreement with a lack of TrkB expression in perirhinal astrocytes, gene deletion of TrkB did not affect LTP induction and maintenance.

Given these data are part of another manuscript that is presently *in preparation*, we will prefer not to include them in our revised manuscript, but rather sharing the results exclusively with the Reviewer#2. **However, if these data will be considered a necessary requisite for the present work, we are ready to incorporate them in the revised version of the present manuscript.**

As astrocytic processes can regulate their distances to synapses and thus contribute to the regulation of synaptic transmission, the proximity of astrocytic processes to synapses could also be modulated following your stimulation of TBS. This could be discussed as well.

Structural remodeling of microdomains might be accompanied by changes in the ability of astrocytes to regulate proBDNF internalization, processing and recycling and ultimately synaptic strengthening. This important issue is now discussed in our revised manuscript.

Point 3. Figure 1 appears to come from the same experiment also illustrated in Figure 1 of Vignoli et al. 2016. Yet, the % co-localization induced by the TBS is much lower. If it is the same experiment, why are these values different? If not, why did the authors change the conditions that would have been more advantageous?

Vignoli et al., (2016) and the present manuscript showed the same types of experiments; however, while in Vignoli et al. (2016) the colocalization was obtained by using α proBDNF (or α mBDNF) antibody, here an α BDNFpro was used. Moreover, in the present manuscript the level of BDNFpro/GFP colocalization signal was normalized to that of the control. This is preferable to better appreciate the overall increase of colocalization in stimulated vs. non-stimulated conditions. In spite of these differences, the extent of colocalization signals that was increased after TBS was comparable in both manuscripts, showing an increment of 2 to 3 times.

Point 4. In the supplementary figure 1, middle and right panels are supposed to be from the same ROI. However, there is more signal on the right panel, that is logically not possible?

This is a clear effect induced by the fact that red signal is less visible. To avoid any misinterpretation of our data by the reader, in Supplementary Fig.1 we have used the same color (white) for both BDNFpro immunoreactivity and BDNFpro/GFP signal. To be consistent we also changed the color of the BDNFpro in Fig. 1b.

Point 5. In Figure 2b ROI n°4, the BDNF pro/GFP signal is not visible. Another illustration should be found.

In Fig. 2b we have now substituted the panel related to the ROI n°4 with a new panel.

Point 6. In the supplementary figure 2c, the histogram should be replaced by individual dot plots. Idem for Fig. 6c and all histograms.

We have now adopted the dot plots graphs in every histogram.

Point 7. No statistics were made to compare the late phases of LTP. Although the effects are clear, statistics should be added.

We have now provided the *p* values related to min 180 post TBS (Fig. 5, Supplementary Fig.4 and Supplementary Fig. 6).

Point 8. In figure 3c, the dashed square for the zoom n°5 is not correctly placed.

Fig. 3c has now been corrected.

Quantifications could also be added to the figure in order to easily access the variability and the quantity of quantified dots.

We added a new panel showing the variability and the quantity of BDNFpro-gold dots in whole astrocytes and peri-synaptic sites (Fig. 3d).

Point 9. P13 line 330 please reformulate “ ... that indicates that”.

We now reformulated the sentence.

Point 10. P14 line 347: I did not understand why the authors measured only pTrkB that colocalized with TrkB and not the total immunoreactivity of neuronal pTrkB. Please clarify.

Given that pTrkB is essentially a fraction of the total receptor levels, we measured in a quantitative analysis pTrkB that co-localizes with TrkB immunoreactivity at the stimulated areas. The benefit for this co-localization is related to the specificity of pTrkB signal: only when pTrkB fluorescence colocalized with TrkB immunoreactivity was the signal considered a phosphorylated receptor. We have clarified this point in the revised version of our manuscript. Moreover, using TrkB immunoreactivity as internal control ensured a better quantification of pTrkB reducing the variability between preparations. In support of our analysis, in Supplementary Fig. 7 we showed TrkB and pTrkB immunoreactivity together with TrkB/pTrkB colocalization signals. To avoid redundancy, we have now removed the same repeated images from Fig. 7.

Point 11. In the discussion section, the signs ==> should be replaced by (i) and (ii), line 405.

We now modified the text as indicated.

Reviewer #3

We thank the Reviewer#3 for the appreciation of our manuscript. Concerning Her/His remarks:

I will comment on the main disputes. The points raised by Rev 1 are valid. Current data and the conclusions offered do not directly demonstrate the proteolytic conversion within astrocytic domains. The authors disagree about the experiments suggested to address the issues, and they do provide a solid explanation about the non-specificity of tPA membrane impermeable inhibition as well as the poor conclusions they will gain from analyses of double stainings with synaptic markers. I agree with the author's arguments.

In agreement with the Reviewer #3 comments, the experiments suggested in points 1, 3 and 4 by Reviewer #1 were not included.

However, the issues remain. The reviewer's points (critiques 1 3 and 4) are valid and it is not the responsibility of the reviewer to identify the best experiments. If the authors know or find a better strategy they should use it. Or they should change the text and indicate more precisely what the data show and point to limitations and alternative explanations. In other words, their conclusions are not supported by direct proof of astrocytic processing. The p75 genetic approach is elegant but does not discriminate between uptake of proBDNF vs. uptake of BDNFpro.

Concerning point #1, we agree that a "direct" demonstration of intracellular proBDNF processing is missing. Biochemical analysis of proBDNF proteolytic processing in astrocytes would require the use of dissociated cultures. However, while in our previous work we employed astrocyte cultures assessing proBDNF internalization and secretion, in the present manuscript we faced the necessity to preserve the integrity of neuron-astrocyte interaction and proBDNF trafficking, as well as to record LTP. Given these constraints, in our original manuscript we provided "indirect" evidence of astrocytic proBDNF processing by a combined use of the specific antibody for the *furin* cleaved C-terminal end of the prodomain in slices from p75-flox mice following TBS.

To reinforce this issue we performed additional experiments and treated our slice preparation with plasmin (100 nM), an enzyme responsible for the extracellular proteolysis of proBDNF to BDNFpro and mBDNF. Concomitant depletion of the precursor and extracellular accumulation of its proteolytic products induced by the enzyme, prevented from both BDNFpro and mBDNF expression in astrocytes (Fig. 1e). Thus, proBDNF, but not BDNFpro and mBDNF, is individually internalized in astrocytes and BDNFpro detection in these cells likely originate from the proBDNF that was previously endocytosed and then cleaved. Accordingly, the uptake of proBDNF in astrocytes was prevented by specific deletion of its carrier receptor p75^{NTR} (Vignoli et al., 2016). Given BDNFpro shows no binding to p75^{NTR} (Anastasia et al., 2013), the lack of BDNFpro expression in p75^{NTR}-deficient

astrocytes (Fig. 1f) imply that the uptake of this isoform from the extracellular space is unlikely to occur.

Concerning point #3 and #4; one way to discriminate among synaptic and astrocytic BDNFpro is to eliminate one of the two components and quantify what is left of the other one. Control experiments in this sense were presented in our original manuscript (Fig. 1e). When BDNFpro internalization was prevented in p75^{NTR}-deficient astrocytes BDNFpro/GFP colocalization was reduced to background levels. Thus, the contribution of synaptic BDNFpro to the overall amount of BDNFpro in the astrocytic territory is minute, if not absent, and did not affect the relevance of our quantification analysis.

In short, if they cannot identify additional experiments to address the points raised by reviewer 1 they certainly can state the reasons why and correct the manuscript accordingly (I would in that case soften the language). The reviewer will judge the responses provided and the manuscript changes and will take it from there. I would allow the authors to resubmit and take the responsibility of what to do, reply, and language changes. Despite our previous and new evidence, a direct biochemical demonstration of proBDNF proteolysis in astrocytes is still missing. As suggested, we have now smoothed the text emphasizing that our findings only “suggest” astrocytic proBDNF processing.

REVIEWERS' COMMENTS:

Reviewer #1 (Remarks to the Author):

In the revised manuscript, authors have adequately responded to all my critique points either by including additional experimental results or by rephrasing and extending explanation/discussion.

From my point of view, manuscript could be accepted for publication in Communications Biology in the present form

Reviewer #2 (Remarks to the Author):

This new version of the manuscript is now improved and convincing. The authors have appropriately responded to all my comments.

Regarding point 2: TrkB invalidation in astrocytes and its impact on LTP induction and maintenance, I would of course prefer the new figure to be in the revised version of the manuscript. However, I understand the authors' point of view and am now convinced by their results, which does not make the presence of this result indispensable in this paper.

In the legends, histograms are often mentioned when they are actually dot plots. Please name the graphs correctly.